# Contribution of task-irrelevant stimuli to drift of neural representations

**Farhad Pashakhanloo**
Center for Brain Science
Harvard University, Cambridge, MA
`fpasha@fas.harvard.edu`

## Abstract

Biological and artificial learners are inherently exposed to a stream of data and experience throughout their lifetimes and must constantly adapt to, learn from, or selectively ignore the ongoing input. Recent findings reveal that, even when the performance remains stable, the underlying neural representations can change gradually over time, a phenomenon known as representational drift. Studying the different sources of data and noise that may contribute to drift is essential for understanding lifelong learning in neural systems. However, a systematic study of drift across architectures and learning rules, and the connection to task, are missing. Here, in an online learning setup, we characterize drift as a function of data distribution, and specifically show that the learning noise induced by task-irrelevant stimuli, which the agent learns to ignore in a given context, can create long-term drift in the representation of task-relevant stimuli. Using theory and simulations, we demonstrate this phenomenon both in Hebbian-based learning—Oja's rule and Similarity Matching—and in stochastic gradient descent applied to autoencoders and a supervised two-layer network. We consistently observe that the drift rate increases with the variance and the dimension of the data in the task-irrelevant subspace. We further show that this yields different qualitative predictions for the geometry and dimension-dependency of drift than those arising from Gaussian synaptic noise. Overall, our study links the structure of stimuli, task, and learning rule to representational drift and could pave the way for using drift as a signal for uncovering underlying computation in the brain.

## 1 Introduction

Continual lifelong learning requires intelligent agents to learn continuously and adaptively from a stream of data and experience [1]. In this process, the internal representations of data may themselves shift or evolve over time. Understanding flexibility and stability of neural representations, as well as the learning algorithms that allow for efficient continual learning is fundamental to both neuroscience and artificial intelligence [2]. In neuroscience, recent advances have allowed for tracking neurons over several weeks or months. Such studies have revealed that representations at the single neuron level might not be as stable as previously thought [3, 4]. This is specifically observed in the context of stable performance and is referred to as representational drift [1] [5–8].

This phenomenon has also been recapitulated in computational models from different perspectives [9–16]. Under one set of postulates, long-term changes in representations are a result of noisy learning [17]. However, it is not clear which sources of noise drives this process. Broadly speaking, this could include biological sources, such as intrinsically noisy synaptic turnover [18] or those related

---

[1]We take representational drift to be any long-term changes in the internal representations that occur after the loss or behavior reaches a steady-state. This applies in both neuroscience and machine learning contexts.

39th Conference on Neural Information Processing Systems (NeurIPS 2025).

to learning from experience [19], such as sampling stochasticity in online learning. Understanding the contributions of different sources of noise could help reverse-engineer mechanisms of learning, especially if each renders a distinct sets of predictions.

In machine learning, a primary source of noise during learning stems from stochastic gradient descent (SGD). There is a large body of work on characterization of SGD noise with a focus on how it can benefit generalization by driving the network toward flatter areas of the loss landscape [20–24]. The noise in SGD has also been shown to drive long-term drift in the network parameters and representations after minimal loss is achieved [24, 25].

It is unclear if drift due to learning noise can also be observed across different architectures and learning rules (including bio-plausible rules), and what are commonalities and differences among these setups. Here, we use theory and simulations to systematically characterize drift as a function of data distribution for different networks and learning rules. We show that certain data-dependency features of drift robustly hold across networks, and carry different predictions than drift caused by other sources of noise.

**Contributions**

- We show in an online learning setup that task-irrelevant data can act as a source of noise, changing the representations of task-relevant data over time despite maintained task performance.

- We analytically study drift in a set of canonical architectures and learning rules, including networks with SGD and Hebbian-based learning. Despite the individual differences, they all exhibit dependency of the drift on task-irrelevant stimuli.

- Using both synthetic and real datasets (MNIST), we show that learning-induced drift leads to different predictions for the geometry and dimension-dependency of the drift than those caused by Gaussian synaptic noise.

## 2   A motivating example: drift under task-irrelevant noise

We start with a simple example to demonstrate the drift induced by task-irrelevant stimuli in a network trained with a Hebbian-based learning. Specifically, we consider a one-layer network trained with multi-dimensional Oja's rule [26, 27]. This is a canonical unsupervised learning method, which learns to represent the principal subspace of data at its output layer. The input to the network is $\boldsymbol{x} \in \mathbb{R}^n$, and the output is given by $\boldsymbol{y} = \boldsymbol{W}\boldsymbol{x} \in \mathbb{R}^m$, where $\boldsymbol{W} \in \mathbb{R}^{m \times n}$ is a trainable weight matrix ($m < n$). The online update rule is:

$$\Delta \boldsymbol{W} = \eta \boldsymbol{y}(\boldsymbol{x} - \boldsymbol{W}^T \boldsymbol{y})^T \quad \text{(Oja's learning rule)}, \tag{1}$$

where $\eta$ is the learning rate. After convergence, the solution weight $\tilde{\boldsymbol{W}}$ aligns with the $m$-principal subspace of data [27]. More concretely, if $\boldsymbol{\Sigma_x} = \boldsymbol{V}\boldsymbol{\Lambda}\boldsymbol{V}^T$ is the singular value decomposition of the input covariance, then $\tilde{\boldsymbol{W}} = \boldsymbol{Q}\boldsymbol{I}_{m,n}\boldsymbol{V}^T$, where $\boldsymbol{Q} \in \mathbb{R}^{m \times m}$ is an orthonormal matrix, and $\boldsymbol{I}_{m,n} \in \mathbb{R}^{m \times n}$ is a rectangular identity matrix with $[\boldsymbol{I}_{m,n}]_{i,j} = \delta_i^j$. We see that the rotational symmetry at the output layer (associated with $\boldsymbol{Q}$) creates a degeneracy for the solution. Here, we demonstrate how online learning, and specifically the statistics of stimuli, could lead to drifting weight matrices within this degenerate space over time. To do so, we consider Gaussian stimuli $\boldsymbol{x} \sim \mathcal{N}(0, \boldsymbol{\Sigma_x})$ with covariance $\boldsymbol{\Sigma_x}$ that has singular values of:

$$\boldsymbol{\Lambda} = \text{diag}([\underbrace{1, .., 1}_{m}, \underbrace{\lambda_\perp, .., \lambda_\perp}_{n-m}]) \tag{2}$$

($\lambda_\perp < 1$). Here, the first $m$ eigenvalues correspond to the $m$-principal subspace of the data $\mathcal{X}_{||}$ that the network learns to represent at its output. Conversely, the complementary $(n-m)$-dimensional subspace $\mathcal{X}_\perp$ associated with eigenvalue $\lambda_\perp < 1$ is repressed at the output (i.e. $\boldsymbol{y} = 0$ for $\boldsymbol{x} \in \mathcal{X}_\perp$). Hence, we call the latter the *task-irrelevant* subspace.

Figure 1 demonstrates simulations of continued online learning in this network after convergence. During this time, the principal subspace is already learned (as measured by a subspace distance), and norms of representations are stationary (Fig. 1a,b). However, the autocorrelation of the representation

of a task-relevant stimulus decays over time, which is compatible with rotational drift. Importantly, we observe that the rate of the autocorrelation decay increases with the variance of the task-irrelevant stimuli ($\lambda_\perp$) as well as the corresponding dimension ($n - m$) (Fig. 1c,d). This is interesting, because it suggests that even though the task-irrelevant stimuli are suppressed at the output, their presence still leads to perturbation and drift of task-relevant stimuli representations over time. To get further

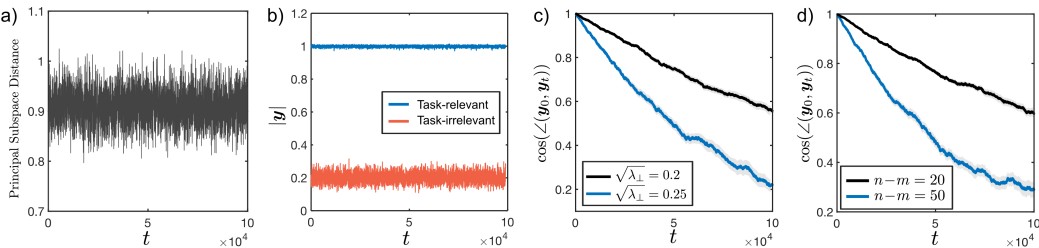

Figure 1: Demonstration of the effect of task-irrelevant stimuli on representational drift. A one-layer network is continuously presented with input samples and updated using Oja's learning rule long after convergence ($n = 50$, $m = 30$, $\eta = 0.025$). a) Grassmann distance between the $m$-principal subspace of the data and row space of $\boldsymbol{W}$. b) Norms of representations for a task-relevant ($\boldsymbol{x}_{||} \in \mathcal{X}_{||}$, blue), and a task-irrelevant stimuli ($\boldsymbol{x}_\perp \in \mathcal{X}_{||}$, orange). Note that, $\boldsymbol{y}_i$'s for $\boldsymbol{x}_\perp$ fluctuate around zero which leads to a non-zero steady-state norm. c) Decay of average cosine similarity for task-relevant stimuli representations, under two values of $\lambda_\perp$. Each curve is averaged for $m = 30$ stimuli; shaded regions indicate the standard error of the mean). d) Same as c) but for $n = 50$ and $n = 80$.

insight into this, let's consider the update equation evaluated at a solution point $\tilde{\boldsymbol{W}}$. By replacing $\boldsymbol{W} = \tilde{\boldsymbol{W}}$ and $\boldsymbol{y} = \tilde{\boldsymbol{W}}\boldsymbol{x}$ in Eq. 1, the instantaneous update after seeing sample $\boldsymbol{x}$ becomes:

$$\Delta \boldsymbol{W}^* = \eta \tilde{\boldsymbol{W}} \boldsymbol{x}_{||} \boldsymbol{x}_\perp^T. \tag{3}$$

In the above, $\boldsymbol{x}_{||}$ and $\boldsymbol{x}_\perp$ are projections of $\boldsymbol{x}$ onto the $m-$principal subspace and the $n - m$ dimensional space orthogonal to that, respectively. This multiplicative term is particularly interesting, as it suggests both components need to be present in a given sample for to drive a deviation from the solution. In the special case where $\boldsymbol{x} \in \mathcal{X}_{||}$, the stimulus already lies within the principal subspace, and no adjustment by the network is necessary. Similarly, when $\boldsymbol{x} \in \mathcal{X}_\perp$, the stimulus is "filtered out" ($\boldsymbol{y} = 0$), and no update occurs. This nonlinear dependence of learning update on stimuli causes task-irrelevant stimuli to act as a source of noise, contributing to long-term dynamics and drift. We will next study this more systematically in the next section.

## 3 Theory and Models

### 3.1 Theoretical approach

We consider a continual (lifelong) learning scenario in which the agent (neural network) experiences stimuli in the environment while performing a particular task. Since drift is experimentally associated with a stable task performance, we assume that the agent has reached its optimal performance and the data are sampled in an online fashion from a stationary distribution, i.e. $\boldsymbol{x} \sim P(\boldsymbol{x})$. (In the supervised learning case, $\boldsymbol{x}$ includes the input-output pair). Let $\boldsymbol{\theta}$ denote the network parameters. After observing sample $\boldsymbol{x}$, a discrete learning update modifies the parameters according to:

$$\Delta \boldsymbol{\theta} = -\eta \boldsymbol{g}(\boldsymbol{x}; \boldsymbol{\theta}), \tag{4}$$

where $\eta$ is the learning rate, and $\boldsymbol{g}(.)$ represents the learning rule. When learning follows gradient descent on an explicit objective, $\boldsymbol{g}$ corresponds to the sample gradient. However, in general, $\boldsymbol{g}$ may represent other update fields, such as Hebbian-based learning. We will consider both cases in the paper. We define the manifold of solutions as a set of parameters that are stable fixed points of the above dynamical system. Previous work has shown that under certain conditions, such as small learning rate, the dynamics of learning with SGD can be approximated using a continuous-time stochastic differential equation (see [24, 25, 28]). We will adopt that approach, and in particular,

similar to [25], decompose the dynamics near a point $\tilde{\boldsymbol{\theta}}$ on the solution manifold into local normal ($N$) and tangential ($T$) spaces to the manifold:

$$\begin{cases} d\boldsymbol{\theta}_N &= -\boldsymbol{H}(\boldsymbol{\theta}_N - \tilde{\boldsymbol{\theta}})dt + \sqrt{\eta}\boldsymbol{C}_N(\boldsymbol{\theta})d\boldsymbol{B}_t \\ d\boldsymbol{\theta}_T &= \sqrt{\eta}\boldsymbol{C}_T(\boldsymbol{\theta})d\boldsymbol{B}'_t. \end{cases} \tag{5}$$

Here, $\boldsymbol{H} = \partial\langle\boldsymbol{g}\rangle_x/\partial\boldsymbol{\theta}|_{\tilde{\boldsymbol{\theta}}}$ denotes the Hessian (or in general the Jacobian of the average dynamics) evaluated at $\tilde{\boldsymbol{\theta}}$, $\boldsymbol{C}_N(\boldsymbol{\theta})$ and $\boldsymbol{C}_T(\boldsymbol{\theta})$ are projections of $\boldsymbol{C}(\boldsymbol{\theta}) = \text{cov}(\boldsymbol{g})^{1/2}$ onto the normal and tangent spaces respectively, and $\boldsymbol{B}_t$ and $\boldsymbol{B}'_t$ denote independent standard Brownian motion. (For clarity, we omit the explicit dependence of $\boldsymbol{g}$ on $\boldsymbol{\theta}$). The above decomposition assumes that $\boldsymbol{H}$ can be block-diagonalized into the normal and tangent spaces. When the learning is gradient descent, this is automatically the case as there exists an explicit loss function, and the projection of $\boldsymbol{H}$ into the tangent space is zero. Among the Hebbian-based learning fields, we found this to hold for Oja's rule [26] and not the Similarity Matching network [29], for which the dynamics are not curl-free. However, we performed a similar decomposition of dynamics using non-orthogonal projections (see Appendix C).

The first equation above describes fast fluctuations orthogonal to the solution manifold. For small perturbations, these dynamics can be approximated by an Ornstein-Uhlenbeck process, from which the corresponding covariance scales as $\propto \eta$. The second equation, governs a diffusion on the manifold itself, which, over long timescales, leads to drift of parameters and representations. Importantly, we focus on a late phase of learning where there is no effective tangential flow along the manifold [11, 24], so the dynamics in the tangential space are purely diffusive. If $\boldsymbol{C}_T(\boldsymbol{\theta})$ does not vanish at solution, i.e. $\boldsymbol{C}_T(\tilde{\boldsymbol{\theta}}) \neq 0$, then the leading-order contribution to diffusion is $D_\theta \propto \eta^2$. Otherwise, one must account for contributions from $\boldsymbol{C}_T(\boldsymbol{\theta})$ at $\boldsymbol{\theta} \neq \tilde{\boldsymbol{\theta}}$. In this case, the effective diffusion also depends on the fluctuations, resulting in scaling of $D_\theta \propto \eta^3$ (see Appendix A for additional details).

## 3.2 Drift across architectures and learning rules

We now apply the above framework to analytically solve for drift in different networks (Figure 2). The choice of networks is made to cover both Hebbian-based and gradient descent learning rules in supervised and unsupervised setups. Importantly, the three unsupervised networks are intrinsically performing the same task of principal subspace tracking but achieving it in different ways. This fact allows us to explore the role of learning rule and architecture in a more controlled way, when the task is the same. Additionally, the two-layer supervised learning setup is more general, allowing for learning an arbitrary linear mapping between the input and the output. However, for comparison to other network, and as a special case, the teacher in that network can also be designed to perform principal subspace tracking. Below, we present the specific models and key results, with a more detailed discussion of Oja's case, as certain aspects of its dynamics are shared with those observed in other networks. Complete derivations are provided in the Appendix. Throughout this section, we assume input data, $\boldsymbol{x} \in \mathbb{R}^n$, has a covariance with eigenvalues in descending order $\lambda_n \leqslant \cdots \leqslant \lambda_2 \leqslant \lambda_1$ [2].

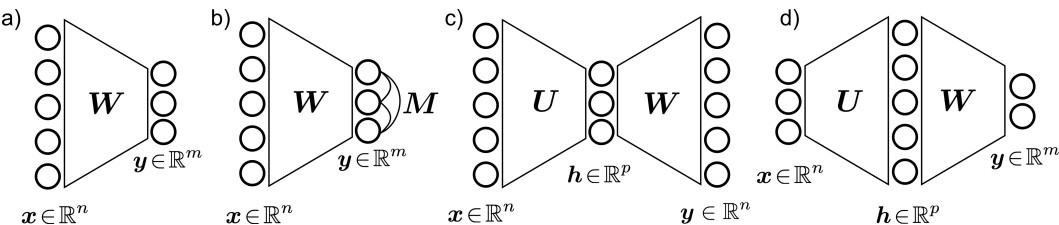

Figure 2: Neural network models studied in this work. a) Multi-dimensional Oja network, b) Similarity Matching network, c) autoencoder with a bottleneck, and d) a two-layer network. The first three networks are trained in an unsupervised way, while the two-layer network is trained with supervised learning.

---

[2]For the networks that perform principal subspace tracking, we assume that the $m$-principal subspace of data is unique. This necessitates a spectral gap between $\lambda_m$ and $\lambda_{m+1}$.

**Multidimensional Oja's network**   As mentioned in Section 2, Oja's network uses a Hebbian-based rule to perform principal subspace tracking at the output [26, 27]. The solution after convergence satisfies $\tilde{W}\tilde{W}^T = I_m$, for $\tilde{W} \in \mathbb{R}^{n \times m}$. This is known as Stiefel manifold and its differential geometry has been characterized previously [30]. An arbitrary deviation from a point $\tilde{W}$ on this manifold can be decomposed as:

$$\delta W = N_1 + N_2 + T \tag{6}$$

$$N_1 = K^{\mu,\nu}\tilde{W}_\perp, \quad N_2 = Z^{i,j}\tilde{W}, \quad T = \Omega^{s,r}\tilde{W},$$

for special matrices $K^{\mu,\nu} \in \mathbb{R}^{m \times (n-m)}$, $Z^{i,j} \in \mathbb{R}^{m \times m}$, skew-symmetric $\Omega^{s,r} \in \mathbb{R}^{m \times m}$, and $\tilde{W}_\perp \in \mathbb{R}^{(n-m) \times n}$ whose row space is orthogonal to that of $\tilde{W}$ ($i,j,\mu,s,r \in [m]$, $\nu \in [n-m]$, see Appendix B). It can be shown that, in the above coordinate system, $H$ becomes diagonal, with positive eigenvalues for $N_1$ and $N_2$ (normal spaces), and zero eigenvalues for $T$ (tangent space).

Further calculations show that the bulk of the fluctuations are in subspace $N_1$ and indeed are induced by task-irrelevant stimuli. This can be seen from the fact that projection of $\Delta W^*$ from Eq. 3 is non-zero along $N_1$, while it vanishes along $N_2$ and $T$. These fluctuations subsequently induce diffusion into the tangent space (see Appendix B). A similar mechanism is observed in other networks.

The relation between the rotational symmetry of representations and the tangent space becomes evident by observing that movement along the $\Omega^{r,s}\tilde{W}$ corresponds to rotations between two orthogonal representations, $y_s = \tilde{W}v_s$ and $y_r = \tilde{W}v_r$, where $v_s$ and $v_r$ are principal axes of the input space. Hence, the diffusion of the entire representation space can be described by $m(m-1)/2$ pairs of angular diffusion coefficients $D_{sr}$ for $r,s \in [m]$, $r > s$ (see [3]). In Appendix B, we show that:

$$D_{sr}^{Oja} = \frac{\eta^3}{16} \sum_{\nu=1}^{n-m} \lambda_{m+\nu}^2 \left( \frac{\lambda_r}{1 - \frac{\lambda_{m+\nu}}{\lambda_r}} + \frac{\lambda_s}{1 - \frac{\lambda_{m+\nu}}{\lambda_s}} \right) \tag{7}$$

In the above, the summation goes through the $(n-m)$-dimensional task-irrelevant space, with the associated variances along different dimensions in that space ($\lambda_{m+\nu}$'s) contributing to the sum. Total diffusion for the representation of a given stimulus, $y_s$ becomes: $D_s = \sum_{r=1, r \neq s}^{m} D_{sr}$.

**Similarity Matching network**   This network optimizes a similarity matching objective using a bio-plausible Hebbian/anti-Hebbian learning rule in a one-layer network with feedforward and recurrent weights ($W \in \mathbb{R}^{m \times n}$, $M \in \mathbb{R}^{m \times m}$ respectively, Figure 2b) [9, 29]. The neuronal dynamics follow the update equation $\tau\dot{y} = Wx - My$, where $\tau$ is much smaller than the time constant of synaptic updates. The synaptic update rules are:

$$\Delta W = \eta(yx^T - W), \quad \Delta M = \eta(yy^T - M) \quad \text{(Similarity Matching learning rule)} \tag{8}$$

The linear version of this network essentially achieves the same principal subspace tracking as Oja's network. However, its update rules are local and hence biologically plausible. The rotational symmetry at the output layer is analogous to that in Oja's case and similarly, it corresponds to infinitesimal changes of the form $\delta F = \Omega^{s,r}\tilde{F}$, where $F = M^{-1}W$ is the filter weight from the input to the output. Consequently, the rotational diffusion can be characterized by the following pairwise coefficients $D_{sr}$, for $r,s \in [m]$, $r > s$ (see Appendix C):

$$D_{sr}^{SM} = \frac{\eta^3}{16\lambda_s\lambda_r} \sum_{\nu=1}^{n-m} \lambda_{m+\nu}^2 \left( \frac{1}{1 - \frac{\lambda_{m+\nu}}{\lambda_r}} + \frac{1}{1 - \frac{\lambda_{m+\nu}}{\lambda_s}} \right) \tag{9}$$

We again see that the summation is over the $(n-m)$-dimensional task-irrelevant space; however, the specific dependence on the spectrum differs from that in Oja's case.

**Autoencoder**   Here we consider a two-layer linear autoencoder with a bottleneck. The encoder and the decoder weights are $U \in \mathbb{R}^{p \times n}$ and $W \in \mathbb{R}^{n \times p}$ respectively, with $p < n$ (Figure 2c). The network is trained with SGD with batch size of one. At the solution, the bottleneck layer represents

---

[3]Because of the diffusive process that underlies representational drift in this work, we will measure diffusion rates as a proxy for drift. Further, as long as drift is associated with rotational transformations of the representation space, it is sufficient to measure pairwise angular diffusion rates $D_{sr}$. This holds for all networks studied here.

the $p-$principal subspace of data [31]. We also assume that the network operates near a balanced weight solution, where the solutions weights $\tilde{U}$ and $\tilde{W}$ satisfy $\tilde{U} = \tilde{W}^T$. The rotational symmetry, in this case, corresponds to coordinated weight changes of the form $\delta W = \Omega^{r,s}\tilde{W}$ and $\delta U = \delta W^T$. Similar to Oja and Similarity Matching cases, pairwise angular diffusion coefficients $D_{sr}$ can be used to characterize the diffusion, which in this case applies to the hidden layer representation $h = Ux$. In Appendix D, we show that for $r, s \in [p]$, $r > s$:

$$D_{sr}^{AE} = \frac{\eta^3 \lambda_r \lambda_s}{64} \sum_{\nu=1}^{n-p} \lambda_{p+\nu}(f(\lambda_s, \lambda_{p+\nu}) + f(\lambda_r, \lambda_{p+\nu})), \tag{10}$$

where $f(\lambda_\mu, \lambda_{p+\nu}) = \sum_{\alpha: Q(\alpha)=0} \frac{\alpha^2}{(1+\alpha^2)(1-\alpha)}$, and $Q(\alpha) = \lambda_{p+\nu}\alpha^2 + (\lambda_\mu - \lambda_{p+\nu})\alpha - \lambda_{p+\nu}$. A common feature of this result with previous cases is the dependence of diffusion on the variance in the task-irrelevant subspace which in the autoencoder case has dimension $n - p$ (note that the diffusion vanishes if all $\lambda_{p+\nu}$ are zero for $\nu \in [n - p]$). The exact dependence on the spectrum is determined by the solutions of the quadratic equation $Q(\alpha) = 0$.

**Two-layer network (supervised)** Finally, we study drift in the hidden layer representation of an expansive two-layer network trained under a linear regression task (Figure 2d). Specifically, the input-output relationship is determined by the teacher signal $y = Px$, where $P \in \mathbb{R}^{m \times n}$ ($m \leqslant n$). This is a more general task than the principal subspace tracking studied for other networks. In this case, the task-irrelevant stimuli lie in the null-space of the mapping $P$ (and for which $y = 0$). We train the network with online SGD and weight decay coefficient $\gamma$. This setup also exhibits a similar rotational symmetry to that observed in the above networks (with most resemblance to the autoencoder network), resulting in rotational diffusion within the hidden layer. We leave the derivation and the results to Appendix E, and only present special results in the next section.

## 4 Numerical results

In this section, we study drift in different experimental setups and datasets using theory and simulations. To measure the drift rate in simulations, we run multiple instances of continued learning, all starting from the same trained network, but undergoing different random data sampling seeds. We then compute the slope of the average autocorrelation curve as an estimate of the drift rate.

### 4.1 Gaussian data

To specifically study the effect of task-irrelevant stimuli to drift, we apply our theoretical predictions to the Gaussian toy data introduced in Eq. 2. To recall, the stimuli covariance eigenvalues are $\lambda_{||} = 1$ in the task-relevant subspace (principal subspace for the unsupervised networks), and are equal to $\lambda_\perp$ in the task-irrelevant. This is a simplified dataset and allows for controlling the effect of task-irrelevant stimuli. Replacing this spectrum in the theoretical predictions for drift in Section 3.2, and assuming $\lambda_\perp \ll 1$ leads to the following results for the total diffusion rates:

$$D_y \approx \frac{\eta^3 \lambda_\perp^2}{8}(m - 1)(n - m) \quad \text{(Oja and Similarity Matching)} \tag{11}$$

$$D_h \approx \frac{\eta^3 \lambda_\perp^2}{32}(p - 1)(n - p) \quad \text{(Autoencoder)} \tag{12}$$

We see that the rate of drift increases with the variance and the dimension in the task-irrelevant space. The dimension of this space for Oja and Similarity Matching is determined by the difference between the input and the output layer dimensions, i.e. $D \propto (n - m)$, while for the autoencoder it depends on the dimension of the input and the bottleneck layer, i.e. $D \propto (n - p)$. The theoretical results show excellent match to drift rates measured in simulation (Figure 3).

For the supervised network, we see a similar dependency of the drift on the task-irrelevant subspace. Specifically, if rank of the input-output mapping $P$ is denoted by $k \leqslant m$, and its non-zero singular values are equal to one, total drift rate for the representation simplifies to (Appendix E):

$$D_h \approx \frac{\eta^3 \gamma^4}{16}(k - 1)(k + 2 + (n - k)\frac{\lambda_\perp}{2}) \quad \text{(Supervised Two-layer)} \tag{13}$$

The above consists of a baseline drift as well as an additional drift term that depends on the dimension and variance of the the task-irrelevant subspace. However, in contrast to previous principal subspace task, the task-irrelevant subspace is determined by the null-space of $\boldsymbol{P}$ and has dimension $n - k$. The principal subspace task can be considered as a special case of the above where the right singular vectors of $\boldsymbol{P}$ align with the principal subspace. Additional simulation results are shown in Figure S1.

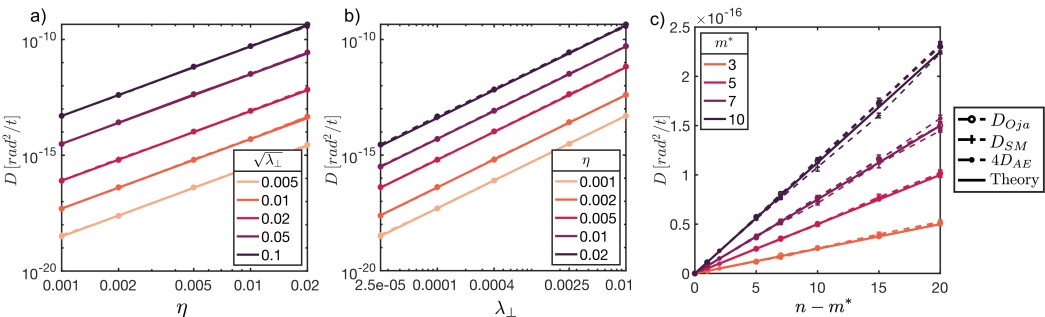

Figure 3: Drift rate for different architectures as a function of: a) learning rate $\eta$, b) task-irrelevant variance $\lambda_\perp$, and c) the dimension of the task-irrelevant space, $n - m^*$. Note that $m^* = m$ for Oja and Similarity Matching (SM), and $m^* = p$ for the autoencoder (AE). For panels a) and b), $n = 5, m = 3$; for panel c), $\eta = 0.001$ and $\sqrt{\lambda_\perp} = 0.01$.

## 4.2 Nonlinear network

We also demonstrate the contribution of task-irrelevant stimuli to drift in a nonlinear network with localized receptive fields. We consider a version of the two-layer network where the network reconstructs the task-relevant position variables on a ring at its output. Specifically, data at the input $\boldsymbol{x} \in \mathbb{R}^n$, consists of task-relevant variables denoted by $x_1 = \cos(\theta)$ and $x_2 = \sin(\theta)$, and task-irrelevant noise represented by $x_i \sim \mathcal{N}(0, \lambda_\perp)$, for $i \in \{3, 4 \ldots n\}$. The output layer has to reconstruct the position, which means the target output is $\boldsymbol{y} = [x_1, x_2]^T$. The predicted output is $\hat{\boldsymbol{y}} = \boldsymbol{W}\boldsymbol{h}$, where $\boldsymbol{h} = \text{ReLU}(\boldsymbol{U}\boldsymbol{x})$ is the activation at the representations layer (Figure 4a). The network is trained with SGD and weight decay, and it learns to represent the position on the ring near perfectly at its output using MSE loss. At this solution, the hidden layer neurons form localized receptive fields (RF) that tile the ring (Figure 4b, similar to [9]). However, continued training leads to reorganization of RFs at the hidden layer, while the kernel (representational similarity matrix) stays stable. Interestingly, an increase in $\lambda_\perp$ leads to a faster decay of representations at the hidden layer (Figure 4c). This suggests that consistent with the results of linear networks, the perturbation created by task-irrelevant stimuli may lead to a lower stability of representations in nonlinear networks.

## 4.3 MNIST data

In the previous section, we used toy Gaussian data which allowed us to simplify the drift equations and study the role of task-irrelevant stimuli in a controlled manner. Here, we apply the theory to MNIST data [32]. Figure 5a, shows two snapshots of representations at the hidden layer of an autoencoder with the bottleneck dimension of $p = 2$ trained on 60000 MNIST images in an online way. We observe an approximate rotation of the representations from one snapshot to another, which is caused by drift of parameters during training. Per the theory in Section 3.2, the drift rate should be a function of the whole spectrum with contributions from the task-irrelevant stimuli. To test this more systematically, we keep the input data the same but change the output/bottleneck dimension for each network. To make the experiments more computationally manageable, we first project the MNIST data onto the top 20 principal components and use that as the input to the network ($n = 20$). Figure 5b shows drift as a function of output for Oja and SM, and the bottleneck dimension for AE. Interestingly, in all cases the drift rate initially increases to a maximum and then decreases to zero at $m = n$ (or $p = n$ for AE), showing great agreement with the theoretical predictions. This trend can be explained by a trade-off between two opposing factors. First, the total space available for drift increases with the dimension of the representation layer (this dependency can also be observed in the Gaussian data in Section 4.1, and it corresponds to factors $m - 1$ in Eq. 11, and $p - 1$ in Eq. 12). The

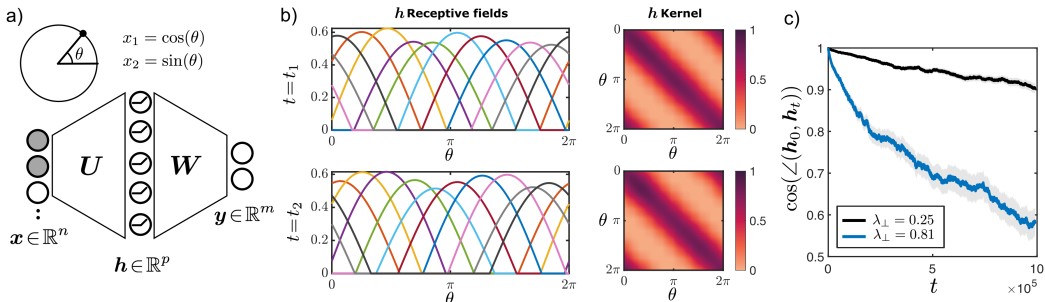

Figure 4: Example of drift in a non-linear network. a) Schematic of a two-layer network trained to represent the position on the ring ($\theta$). b) Reorganization of receptive fields after continued training, as shown by two snapshots $t_1 = 1.25{\times}10^6$ and $t_2 = 5{\times}10^6$. The right panel shows the stability of kernel $K(\theta_1, \theta_2) = \boldsymbol{h}(\theta_1)^T \boldsymbol{h}(\theta_2)$ across time. c) Decay of average cosine similarity and its dependence on $\lambda_\perp$ (averages were performed over representations of 5 uniformly chosen angles and 80 runs, shaded area: standard error of the mean). Unless otherwise specified, for all simulations: $n = 8$, $p = 10$, $m = 2$, $\eta = 0.2$, $\gamma = 0.05$ and $\lambda_\perp = 0.25$.

second factor can be attributed to the source of noise that exists during learning, which as we saw is driven by the task-irrelevant stimuli. As the output or the bottleneck dimensions increase, this space effectively shrinks in size, reducing the amount of noise and thereby decreasing the drift. This is why, at the limit where this dimension equals the input dimension, the drift vanishes altogether.

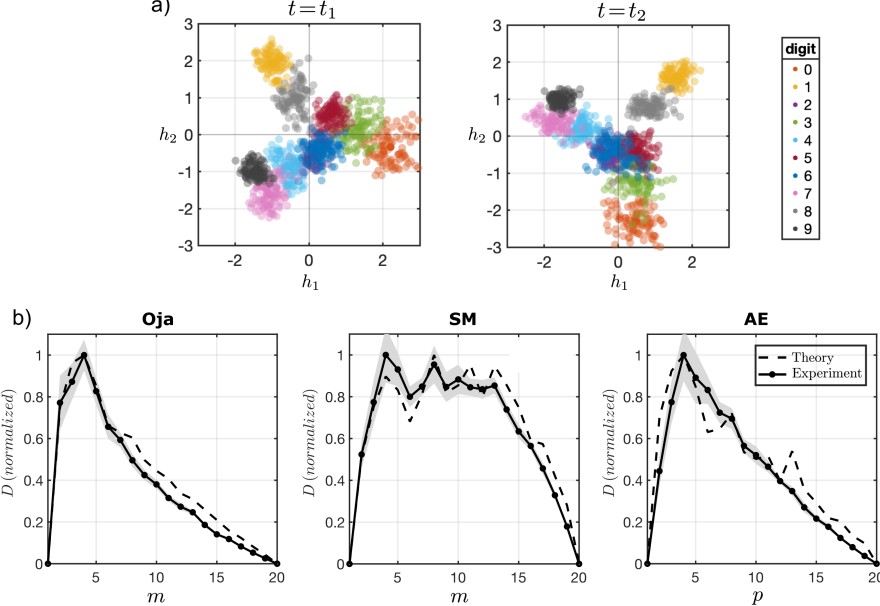

Figure 5: Drift in MNIST data. a) Two snapshots of hidden layer representations for 10 sample digits in a linear autoencoder ($n = m = 784$, $p = 2$, $\eta = 0.01$, $t_1 = 1.6{\times}10^4$ and $t_2 = 10^5$). To show the fluctuations alongside the drift, a time window of $t_w = 5000$ preceding each snapshot is considered and representations at 100 uniformly chosen times are overlaid. b) Normalized rate of drift as a function of output dimension ($m$) for Oja and Similarity Matching (SM), and bottleneck dimension ($p$) for autoencoder (AE). For the results in this panel, projections of MNIST data onto its top $n = 20$ principal components are used as input. The plots show the drift rate for the representation of the top principal vector (shaded area: standard deviation over multiple runs).

.

# 5 Learning noise vs. synaptic noise

So far, we have shown that noise due to online learning is sufficient to create drift, even in the absence of explicit synaptic noise. A natural question arises as to how these two types of noise lead to different qualitative and quantitative effects on drift. To explore this question, consider a case where in addition to the intrinsic learning noise, an additive synaptic noise is present during learning [9]. In the case of Oja's network, the noisy update becomes:

$$\Delta \boldsymbol{W} = \eta \boldsymbol{y}(\boldsymbol{x} - \boldsymbol{W}^T \boldsymbol{y})^T + \boldsymbol{\varepsilon}, \tag{14}$$

where $\varepsilon_{ij} \sim \mathcal{N}(0, \eta \sigma_{syn}^2)$ is additive Gaussian synaptic noise, and $\sigma_{syn}$ determines its strength. Unlike what we previously observed in the case of pure learning noise, the projection of synaptic noise to the tangent space of the solution manifold is non-zero. This makes the calculation of the diffusion on the manifold simpler than the case of learning noise. Specifically, if $\boldsymbol{T}$ is an arbitrary unit vector in the tangent space, we have:

$$\langle \mathrm{proj}(\boldsymbol{\varepsilon}, \boldsymbol{T})^2 \rangle_{noise} = \eta \sigma_{syn}^2. \tag{15}$$

By ignoring higher order terms, the perturbations caused by synaptic noise along the tangent space accumulate over time while those in other directions are mean-reverted to zero. It is easy to show that the corresponding pairwise diffusion between two orthogonal representations is $D_{sr} = \eta \sigma_{syn}^2 / 4$, where one factor of $1/2$ results from conversion from the parameter to the representation space, and another $1/2$ from the definition of diffusion. A close comparison of this to the pairwise drift caused by intrinsic learning noise reveals that the synaptic-induced diffusion is isotropic while the learning-induced drift is in general anisotropic (the latter can be observed from Eq. 7 where for a given stimulus, the drift rate of representation along different directions is not the same). Hence, the "geometry" of drift shows a major qualitative difference between the two sources of noise. Additionally, we may also compare the overall drift rates between these two sources of noise. For the Gaussian toy data in Section 4.1, total diffusion for a representation becomes:

$$D_y \approx \frac{\eta^3 \lambda_\perp^2}{8}(m-1)(n-m) + \frac{1}{4}\eta \sigma_{syn}^2 (m-1), \tag{16}$$

where the first term denotes the drift caused by intrinsic learning noise, and the second term stems from additive synaptic noise. The overall drift rate is plotted in Figure 6a as a function of the synaptic noise strength for the Gaussian data. We see that synaptic noise dominates when the strength is higher than $\sigma_{syn}^* \sim \eta \lambda_\perp \sqrt{n-m}$. Additionally, as expected, the drift rate does not vanish at small values of $\sigma_{syn}$; instead it plateaus at a level that depends on the task-irrelevant noise.

Another qualitative difference between the two sources of noise is evident from the dependency of the drift on the output dimension. In the previous section where the only source of noise was learning noise, we showed that the drift rate may have a non-monotonic relationship with the output dimension. Here, we repeat the MNIST experiments with different levels of synaptic noise during learning. As shown in Figure 6b, as the strength of synaptic noise increases, the relationship approaches a monotonically increasing function of the output dimension, yielding a qualitatively different dependency.

# 6 Discussion

Here, we applied a theoretical framework to study representational drift across a set of canonical networks and learning rules. Our focus was on understanding drift arising from the inherent stochasticity of online learning. A key finding was that task-irrelevant stimuli can act as a source of fluctuation during learning, potentially leading to long-term drift in the representations. This phenomenon was consistently observed across all the networks studied, and rendered different predictions than an additive synaptic noise.

One feature of the chosen networks was that they perform similar tasks, but differ in their architectures and learning rules. The exact dependency of the drift on the data distribution was different for each architecture. Nevertheless, in all cases, the extent of stimuli that the network learns to ignore influenced the drift of representations of task-relevant stimuli. At first, this might sound unintuitive; however, the online nature of learning prevents the networks from being fully agnostic to a part of the data distribution. Such sensitivity allows for adaptation in the case of non-stationary data.

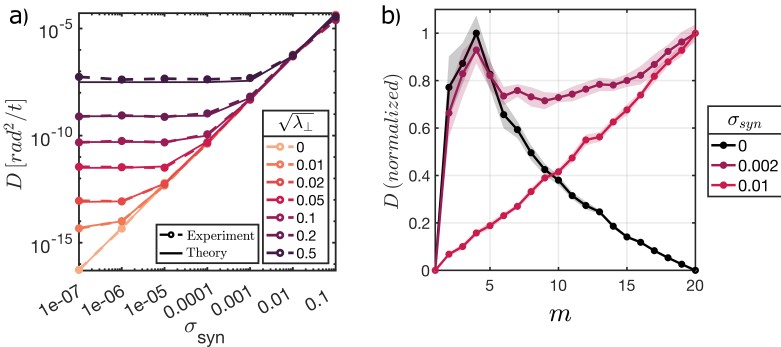

Figure 6: Comparison of drift induced by learning noise from task-irrelevant data and by Gaussian synaptic noise in Oja's network. a) Drift rate as a function of synaptic noise strength ($\sigma_{syn}$), shown for different values of $\lambda_\perp$ in the Gaussian data ($n = 5$, $m = 3$, $\eta = 0.01$). b) Normalized drift rate for MNIST data as a function of output dimension ($m$), shown for three values of $\sigma_{syn}$. Curves are normalized to their maximum values separately (shaded area: standard deviation over multiple runs).

This may have some resemblance to the problem of ongoing memory storage [14]. In the case of gradient-based methods where the loss function is explicit (AE and two-layer networks in our work), this can be explained by non-vanishing sample loss which leads to persistent parameter updates even after convergence. Previous work has shown that SGD with weight decay in a two-layer autoencoder with an expansive hidden layer can induce drift [25]. Our work extends those findings to different architectures and to supervised learning, and highlights the role of task-irrelevant stimuli as a distinct source of instability.

Drift has been observed across various cortical areas and under different experimental designs. Nevertheless, we did not find any studies that specifically examined the influence of task-irrelevant stimuli. We can, however, speculate about potential connections to some existing findings. In [33], the amount of drift observed in an association area—where different types of stimuli are multiplexed—appears to be higher than drift at the origin areas. This might be related to the phenomenon discussed in our study as different sets of stimuli are combined with varying levels of relevance depending on the task. Additionally, a potentially related observation comes from the visual cortex, where more complex and naturalistic stimuli show higher drift compared to simpler stimuli [34]. This could also relate to our finding that, in a network with a limited representational capacity (such as the bottleneck in our work), a more complex stimulus activates a higher number of input eigenmodes, resulting in a greater amount of "task-irrelevant" content, and consequently a higher drift. Finally, in our work all stimuli may be simultaneously present and sampled in the same environment. We did not consider cases where the task-irrelevant stimuli appear in a separate context as studied in [35].

In this work, the theoretical derivations of drift were limited to linear networks. These networks, despite the linearity of the input-output relationship, have been shown to demonstrate nonlinear learning dynamics [36]. This indeed underlies many of the findings in our work, namely the dependency of drift on task-irrelevant stimuli, which was observed in both SGD and Hebbian learning. Along this line, we also found rather complicated functional dependency of the drift on the data spectrum, the specifics of which varied for each architecture and only simplified under structured spectra. Nevertheless, we showed numerically that our main finding also holds for nonlinear activation. Additionally, our theoretical framework relies on approximations of small learning rates, and fluctuations around the solution. For large learning rates, effects such as the finite step-size effect or edge of stability may come into play and lead to more complicated dynamics[37, 38].

Noise due to sample stochasticity, and specifically task-irrelevant stimuli, might be among many sources of noise simultaneously present in the brain [17]. Here, we showed that in the regime where Gaussian synaptic noise is strong and dominates, the rate of drift increases with the dimension of the representation layer, while a more nuanced relationship exists for the sample-induced noise. These different predictions could help guide experiments aimed at uncovering the sources of noise and nature of drift [39]. Specifically, this could be tested in future experiments where the amount of task-irrelevant stimuli is controllable by the experimenter, such as in olfactory figure-ground segregation tasks in which the number and the extent of distracting stimuli can be controlled systematically [40]. Our work lays a theoretical foundation for those endeavors.

## Acknowledgments

I would like to thank Jacob Zavatone-Veth for helpful discussion and comments on the manuscript. I am also grateful to Juan Carlos Fernández del Castillo and Venkatesh Murthy for their feedback on an earlier version of this manuscript. This work was supported by the Harvard Center for Brain Science (CBS)-NTT Fellowship Program on the Physics of Intelligence. The computations in this paper were run on the FASRC Cannon cluster supported by the FAS Division of Science Research Computing Group at Harvard University.

## Code Availability

The codes and core functions to reproduce the main results are publicly available at: https://github.com/fpashakhanloo/task-irrel-drift.

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

## Appendix

## A  Summary of Framework

Here, we provide a summary of the framework, and specifically provide additional details on deriving the fluctuations and diffusion dynamics, which are necessary for solving specific networks in later sections. The decomposition of dynamics is based on the framework in Ref. [25] but we extend that work beyond SGD.

### A.1  Fluctuations

For small learning rates, the stochastic differential equation (SDE) for the deviations in the normal space (first line in Eq. 5), can be approximated as the following Ornstein Uhlenbeck (OU) process:

$$d\boldsymbol{\theta}_N = -\boldsymbol{H}\boldsymbol{\theta}_N dt + \sqrt{\eta}\,\boldsymbol{C}\,d\boldsymbol{B}_t. \tag{17}$$

Here, we replaced $\boldsymbol{C}(\boldsymbol{\theta})$ in Eq. 5 with its value on the manifold. This means $\boldsymbol{C} = \boldsymbol{C}(\tilde{\boldsymbol{\theta}}) = \langle \boldsymbol{g}_* \boldsymbol{g}_*^T \rangle_x^{1/2}$, where $\boldsymbol{g}_*(x)$ is the sample update on the manifold. We define coordinates $\rho$'s within the normal space as below:

$$\boldsymbol{\rho} = \boldsymbol{N}^T \boldsymbol{\theta}_N, \quad \text{where } \boldsymbol{N} = [\boldsymbol{n}_1 | \boldsymbol{n}_2 | ... | \boldsymbol{n}_K], \tag{18}$$
$$\text{and } \boldsymbol{H}\boldsymbol{n}_k = \lambda_k \boldsymbol{n}_k, \ \lambda_k > 0.$$

The stationary correlation function in these coordinates can be derived by solving the above OU process:

$$\langle \rho_k(t)\rho_l(s) \rangle = \frac{\eta \langle \boldsymbol{n}_k^T \boldsymbol{g}_* \, \boldsymbol{n}_l^T \boldsymbol{g}_* \rangle_x}{\lambda_k + \lambda_l} e^{-\lambda_k(t-s)}, \quad (t \geqslant s, \ k,l \in [K]). \tag{19}$$

(see [25, 41]). And finally, the stationary covariance can be obtained from the above by setting $t = s$:

$$\langle \rho_k \rho_l \rangle = \frac{\eta}{\lambda_k + \lambda_l} \langle \boldsymbol{n}_k^T \boldsymbol{g}_* \, \boldsymbol{n}_l^T \boldsymbol{g}_* \rangle_x, \quad k,l \in [K]. \tag{20}$$

To calculate the above terms, one needs to find the normal eigenvectors $\boldsymbol{n}_k$'s, as well as the projection of learning update on them. We use the above equation to find fluctuation variance for all the learning cases in future sections.

### A.2  Diffusion

In this section, we approximate the learning dynamics on the manifold to derive effective diffusion coefficients for the representations. The discrete update along the manifold, evaluated at point $\boldsymbol{\theta} = \tilde{\boldsymbol{\theta}} + \boldsymbol{N}\boldsymbol{\rho}$, can be written as:

$$\Delta \tilde{\boldsymbol{\theta}} = -\eta \boldsymbol{g}_T(\boldsymbol{x}; \tilde{\boldsymbol{\theta}} + \boldsymbol{N}\boldsymbol{\rho}), \tag{21}$$

where $\boldsymbol{g}_T = \Pi_T(\boldsymbol{g})$ is the projection of the update vector onto the tangent space (note that, if $\boldsymbol{H}$ block-diagonalizes in the tangent and normal spaces, this projection is a Euclidean projection. Otherwise, this can be a non-orthogonal projection, as is the case for the Similarity Matching shown in Appendix C). The value of $\boldsymbol{g}_T$ is calculated at a point away from the manifold, and hence it depends on $\boldsymbol{\rho}$. In the absence of any flow in the tangent space, the displacement along the manifold follows an effective diffusion process, with the diffusion tensor of $\boldsymbol{D}_\theta \propto \langle \boldsymbol{g}_T \boldsymbol{g}_T^T \rangle_{x,\rho}$ in the parameter space.

Since we are interested in rotational drift in the representation space, we can define corresponding angular diffusion coefficients that characterize the rotational diffusion in that space. Specifically, we

define $D_{sr}$ to be the angular diffusion rate between the representations $\boldsymbol{h}_s$ and $\boldsymbol{h}_r$ (or alternatively, $\boldsymbol{y}_s$ and $\boldsymbol{y}_r$ if we consider the output layer as the representation layer, which was the case for Oja and SM). To find those coefficients, we need to project $\boldsymbol{D}_\theta$ along the corresponding axis of the tangent space and make appropriate conversion from parameter to representation space. By expanding Eq. 21 to the leading term in $\rho$ (assuming small fluctuations), the pairwise diffusion coefficients become ([25]):

$$D_{sr} := \frac{1}{2}\langle \Delta\varphi_{sr}^2 \rangle_{x,\rho} = \frac{\eta^2}{2}\sum_{k,l=1}^{K}\langle\rho_k\rho_l\rangle\langle\mathcal{G}_k^{s,r}\mathcal{G}_l^{s,r}\rangle_x. \tag{22}$$

Here, coefficients $\mathcal{G}_l^{s,r}(\boldsymbol{x})$ characterize the amount of angular perturbation between $\boldsymbol{h}_s$ and $\boldsymbol{h}_r$ that results from observing sample $\boldsymbol{x}$, when deviation is along $\boldsymbol{n}_k$; these can be calculated for each specific network. It should be noted that, in general, if $\boldsymbol{g}_T$ does not vanish on the manifold, we should also have zeroth order terms in the above. However, we did not observe that in any of the networks studied.

## B  Oja Derivations

Recall from the main text that, if $\boldsymbol{\Sigma_x} = \boldsymbol{V\Lambda V}^T$ is the singular value decomposition of the input covariance, the solution satisfies:

$$\tilde{\boldsymbol{W}} = \boldsymbol{Q}\boldsymbol{I}_{m,n}\boldsymbol{V}^T. \tag{23}$$

Here, $\boldsymbol{Q} \in \mathbb{R}^{m\times m}$ is an orthonormal matrix, and $\boldsymbol{I}_{m,n} \in \mathbb{R}^{m\times n}$ is a rectangular identity matrix, i.e. $[\boldsymbol{I}_{m,n}]_{i,j} = \delta_i^j$. We denote the eigenvalues of $\boldsymbol{\Sigma_x}$ by $\lambda_i$'s. This solution, as well as its stability, has been shown previously (see [27]).

### B.1  Average dynamics near the fixed points

The average flow near the solution manifold can be obtained by taking the expectation of the update rule $\Delta\boldsymbol{W} = \eta\boldsymbol{y}(\boldsymbol{x} - \boldsymbol{W}^T\boldsymbol{y})^T$ with respect to $\boldsymbol{x}$:

$$\langle\Delta\boldsymbol{W}\rangle_x = \eta\boldsymbol{W}\langle\boldsymbol{xx}^T\rangle_x(\boldsymbol{I}_n - \boldsymbol{W}^T\boldsymbol{W}). \tag{24}$$

Computing the above at a point $\tilde{\boldsymbol{W}} + \delta\boldsymbol{W}$ near the manifold, and keeping the leading terms yields:

$$\langle\Delta\boldsymbol{W}\rangle_x|_{\tilde{\boldsymbol{W}}+\delta\boldsymbol{W}} = \eta\delta\boldsymbol{W}\boldsymbol{\Sigma_x}(\boldsymbol{I}_n - \tilde{\boldsymbol{W}}^T\tilde{\boldsymbol{W}}) - \eta\tilde{\boldsymbol{W}}\boldsymbol{\Sigma_x}(\tilde{\boldsymbol{W}}^T\delta\boldsymbol{W} + \delta\boldsymbol{W}^T\tilde{\boldsymbol{W}}) + \mathcal{O}(|\delta\boldsymbol{W}|^2). \tag{25}$$

The eigenvalue equation for $\boldsymbol{H}$, the Jacobian of the dynamics (see Eq. 5) satisfies $\boldsymbol{H}\text{vec}(\boldsymbol{N}) = \lambda_H\text{vec}(\boldsymbol{N})$, where $\boldsymbol{N} \in \mathbb{R}^{m\times n}$ and $\lambda_H$ are the eigenvectors and the eigenvalues respectively, and the "vec" operator reshapes a matrix to a vector. In terms of the $\boldsymbol{N}$ matrix directly, this can be written as:

$$-\boldsymbol{N}\boldsymbol{\Sigma_x}(\boldsymbol{I}_n - \tilde{\boldsymbol{W}}^T\tilde{\boldsymbol{W}}) + \tilde{\boldsymbol{W}}\boldsymbol{\Sigma_x}(\tilde{\boldsymbol{W}}^T\boldsymbol{N} + \boldsymbol{N}^T\tilde{\boldsymbol{W}}) = \lambda_H\boldsymbol{N} \tag{26}$$

As mentioned in the main text, an arbitrary deviation from the manifold can be described as ([30]):

$$\delta\boldsymbol{W} = \boldsymbol{N}_1 + \boldsymbol{N}_2 + \boldsymbol{T} \tag{27}$$

$$\boldsymbol{N}_1 = \boldsymbol{K}^{\mu,\nu}\tilde{\boldsymbol{W}}_\perp, \quad \boldsymbol{N}_2 = \boldsymbol{Z}^{i,j}\tilde{\boldsymbol{W}}, \quad \boldsymbol{T} = \boldsymbol{\Omega}^{s,r}\tilde{\boldsymbol{W}}.$$

One can show that the $\boldsymbol{H}$ is indeed diagonalized in the above coordinates. Below, we will mention the exact coordinates corresponding to each subspace, as well as the associated $\lambda_H$. It is straightforward to show that each of the subspaces satisfies Eq. 26.

**Subspace $\boldsymbol{N}_1$ :**

$$\boldsymbol{N}_1 = \boldsymbol{K}^{\mu,\nu}\tilde{\boldsymbol{W}}_\perp, \quad \text{where } \boldsymbol{K}^{\mu,\nu} \in \mathbb{R}^{m\times(n-m)}, [\boldsymbol{K}^{\mu,\nu}]_{ij} = q_{i\mu}\delta_j^\nu, \quad \mu \in [m], \nu \in [n-m]$$

$$\lambda_H^{\mu,\nu} = \lambda_\mu - \lambda_{m+\nu}, \qquad dim(\boldsymbol{N}_1) = m(n-m)$$

In the above, $\tilde{\boldsymbol{W}}_\perp \in \mathbb{R}^{(n-m)\times n}$ is a full-rank matrix whose row space is orthogonal to that of $\tilde{\boldsymbol{W}}$. Also, recall that $\boldsymbol{q}_i$'s are the columns of the orthonormal matrix $\boldsymbol{Q}$ in 23. The interpretation of displacements in this normal subspace is that the representations of $\boldsymbol{x}_{m+\nu} \in \mathcal{X}_\perp$, which are on average zero, move toward representations of $\boldsymbol{x}_\mu \in \mathcal{X}_{||}$.

**Subspace $N_2$:**

$$N_2 = Z^{i,j}\tilde{W}, \quad Z^{i,j} = \frac{\lambda_i}{\lambda_j}q_i q_j^T + q_j q_i^T, \quad i,j \in [m], i \geq j,$$

$$\lambda_H^{i,j} = \lambda_i + \lambda_j \qquad dim(N_2) = \frac{m(m+1)}{2}.$$

**Subspace $T$:**

$$T = \Omega^{s,r}\tilde{W}, \quad \Omega^{s,r} = \frac{1}{\sqrt{2}}(q_r q_s^T - q_s q_r^T), \quad r,s \in [m]$$

$$\lambda_H^{s,r} = 0, \qquad dim(T) = \frac{m(m-1)}{2}$$

This is a tangential subspace and it corresponds to rotations within the m-dimensional output space.

## B.2 Fluctuations

Update on the manifold can be derived by replacing $W = \tilde{W}$ in the Oja's update equation (Eq. 1):

$$\Delta W_* = \eta y(x - \tilde{W}^T y)^T \tag{28}$$
$$= \eta \tilde{W} x(x^T - x^T \tilde{W} \tilde{W})$$
$$= \eta \tilde{W} xx^T(I_n - \tilde{W}^T \tilde{W})$$
$$= \eta \tilde{W} x_{||}x_\perp^T$$

In the above, $x_{||}$ and $x_\perp$ are projections of $x$ onto the $m-$principal subspace and the $n - m$ dimensional subspace orthogonal to that, respectively. This is the same equation as Eq. 3 in the main text.

Next we observe that among the above three subspaces, the projection of $\Delta W_*$ is only non-zero along $N_1$. This can be easily verified by using the equation of inner product between two matrices, namely for $A, B \in \mathbb{R}^{m \times n}$, the inner product is: $\langle A, B \rangle = \text{tr}(A^T B)$. The projection onto $N_1$ becomes:

$$\text{proj}(\Delta W_*, K^{\mu,\nu}\tilde{W}_\perp) = \eta x_\mu x_{m+\nu}, \tag{29}$$

where we defined $\text{proj}(a, b) := \frac{b^T a}{\|b\|}$. Correspondingly, $\text{proj}(\Delta W_*, N_2) = 0$ and $\text{proj}(\Delta W_*, T) = 0$. Here, $x_\mu = v_\mu^T x$ is a component of stimulus in the principal subspace, and $x_\nu = v_{m+\nu}^T x$ a component in the task-irrelevant subspace. Replacing the above and the corresponding $\lambda_H^{\mu,\nu}$ in Eq. 20, we obtain the covariance of fluctuations associated with this subspace:

$$\langle \rho_{\mu,\nu}^2 \rangle = \frac{\eta \langle x_\mu^2 x_{m+\nu}^2 \rangle}{2(\lambda_\mu - \lambda_{m+\nu})} = \frac{\eta \lambda_{m+\nu}}{2(1 - \frac{\lambda_{m+\nu}}{\lambda_\mu})} \qquad \mu \in [m], \nu \in [n - m], \tag{30}$$

and the other cross-terms are zero. The above indicates the extent to which the representation vector $h_{m+\nu}$ fluctuates toward $h_\mu$.

## B.3 Calculating the Diffusion

We already saw in the above that the only fluctuations occur in the subspace $N_1$. We will next approximate the gradient near the manifold at a point $W = \tilde{W} + \rho K \tilde{W}_\perp$, where $\rho$ indicates the extent of deviation:

$$\Delta W|_{\tilde{W}+\rho K \tilde{W}_\perp} = \eta W xx^T(I_n - W^T W) \tag{31}$$
$$= \eta(\tilde{W} + \rho K \tilde{W}_\perp)xx^T(I_n - (\tilde{W} + \rho K \tilde{W}_\perp)^T(\tilde{W} + \rho K \tilde{W}_\perp))$$
$$= \eta \tilde{W} xx^T(I_n - \tilde{W}^T \tilde{W})$$
$$+ \eta\rho[K W_\perp xx^T(I_n - \tilde{W}^T \tilde{W}) - \tilde{W} xx^T W_\perp^T K^T \tilde{W} - \tilde{W} xx^T \tilde{W}^T K W_\perp] + \mathcal{O}(\rho^2)$$

The projection of the above onto the tangent space $\boldsymbol{T} = \boldsymbol{\Omega}^{s,r}\tilde{\boldsymbol{W}}$ becomes:

$$\text{proj}(\Delta\boldsymbol{W}|_{\tilde{\boldsymbol{W}}+\rho\boldsymbol{K}^{\mu,\nu}\tilde{\boldsymbol{W}}_\perp}, \boldsymbol{\Omega}^{s,r}\tilde{\boldsymbol{W}}) = \frac{1}{\sqrt{2}}\eta\rho x_{m+\nu}(\delta_s^\mu x_r - \delta_r^\mu x_s). \tag{32}$$

We can now use Eq. 22 to calculate the pairwise diffusion constants. In that formulation, we have the following tensor coefficients:

$$\mathcal{G}_{\mu,\nu}^{s,r} = \frac{1}{2}x_{m+\nu}(\delta_s^\mu x_r - \delta_r^\mu x_s), \quad \mu \in [m], \nu \in [n-m] \tag{33}$$

Recall that this coefficient characterizes the extent of displacement between representations of stimuli $\boldsymbol{v}_s$ and $\boldsymbol{v}_r$ upon observing sample $\boldsymbol{x}$, when the network is deviated from the solution along $\boldsymbol{n}_1^{\mu,\nu}$. The corresponding pairwise diffusion coefficients become:

$$D_{sr} = \frac{\eta^3}{2}\sum_{\mu\in[m]}\sum_{\nu\in[n-m]}\langle\rho_{\mu,\nu}^2\rangle\langle(\mathcal{G}_{\mu,\nu}^{s,r})^2\rangle_x \qquad s,r \in [m] \tag{34}$$

$$= \frac{\eta^3}{8}\sum_{\nu\in[n-m]}\langle x_{m+\nu}^2\rangle(\langle\rho_{r,\nu}^2\rangle\langle x_r^2\rangle + \langle\rho_{s,\nu}^2\rangle\langle x_s^2\rangle)$$

$$= \frac{\eta^3}{16}\sum_{\nu\in[n-m]}\lambda_{m+\nu}^2\left(\frac{\lambda_r}{1-\frac{\lambda_{m+\nu}}{\lambda_r}} + \frac{\lambda_s}{1-\frac{\lambda_{m+\nu}}{\lambda_s}}\right).$$

In the last line we replaced the fluctuation covariance terms $\langle\rho_{\mu,\nu}^2\rangle$ by its value from Eq. 30.

## C  Similarity Matching

As discussed in the main text, the network includes a feedforward weight $\boldsymbol{W} \in \mathbb{R}^{m\times n}$, and a recurrent weight $\boldsymbol{M} \in \mathbb{R}^{m\times m}$. The input-output relationship could be represented as $\boldsymbol{y} = \tilde{\boldsymbol{F}}\boldsymbol{x}$, where $\tilde{\boldsymbol{F}} = \tilde{\boldsymbol{M}}^{-1}\tilde{\boldsymbol{W}}$ is the filter weight matrix. Similar to Oja, at the solution (fixed points), the network learns to represent the principal subspace of the input at the output [29]. If $\boldsymbol{\Sigma_x} = \boldsymbol{V}\boldsymbol{\Lambda}\boldsymbol{V}^T$ is the SVD of the data covariance, the solutions satisfy:

$$\tilde{\boldsymbol{M}} = \boldsymbol{Q}\boldsymbol{I}_{m,n}\boldsymbol{\Lambda}\boldsymbol{I}_{m,n}^T\boldsymbol{Q}^T, \quad \tilde{\boldsymbol{W}} = \boldsymbol{Q}\boldsymbol{I}_{m,n}\boldsymbol{\Lambda}\boldsymbol{V}^T, \quad \tilde{\boldsymbol{F}} = \boldsymbol{Q}\boldsymbol{I}_{m,n}\boldsymbol{V}^T, \tag{35}$$

where $\boldsymbol{I}_{m,n} \in \mathbb{R}^{m\times n}$ is a rectangular identity matrix with $[\boldsymbol{I}_{m,n}]_{i,j} = \delta_i^j$, and $\boldsymbol{Q} \in \mathbb{R}^{m\times m}$ is an orthonormal matrix that accounts for the rotational degeneracy of the network. Stability of this network has been previously studied in [29].

### C.1  Average dynamics near fixed points

The online update rule is:

$$\begin{cases} \Delta\boldsymbol{W} = \eta(\boldsymbol{y}\boldsymbol{x}^T - \boldsymbol{W}) \\ \Delta\boldsymbol{M} = \eta(\boldsymbol{y}\boldsymbol{y}^T - \boldsymbol{M}) \end{cases} \tag{36}$$

Similar to [29], we find it convenient to change coordinates to $\boldsymbol{\theta} = (\boldsymbol{F}, \boldsymbol{M})$. The average update at a point $\tilde{\boldsymbol{\theta}} + \boldsymbol{n} = (\tilde{\boldsymbol{F}} + \boldsymbol{N}_F, \tilde{\boldsymbol{M}} + \boldsymbol{N}_M)$ near the manifold of solutions becomes:

$$\begin{cases} \langle\Delta F\rangle_x|_{\tilde{\boldsymbol{\theta}}+\boldsymbol{n}} \approx \tilde{\boldsymbol{M}}^{-1}\boldsymbol{N}_F\boldsymbol{\Sigma_x}(\boldsymbol{I}_n - \tilde{\boldsymbol{F}}^T\tilde{\boldsymbol{F}}) - \boldsymbol{N}_F - \tilde{\boldsymbol{F}}\boldsymbol{N}_F^T\tilde{\boldsymbol{F}} \\ \langle\Delta M\rangle_x|_{\tilde{\boldsymbol{\theta}}+\boldsymbol{n}} \approx \boldsymbol{N}_F\boldsymbol{\Sigma_x}\tilde{\boldsymbol{F}}^T + \tilde{\boldsymbol{F}}\boldsymbol{\Sigma_x}\boldsymbol{N}_F^T - \boldsymbol{N}_M \end{cases} \tag{37}$$

where the averages are performed with respect to samples, and terms of $\mathcal{O}(|\boldsymbol{n}|^2)$ and higher are ignored. The right side of the above equations can be written in terms of the Jacobian of the average dynamics, $\boldsymbol{H}$. The eigenvalue equation for the Jacobian is $\boldsymbol{H}\boldsymbol{n} = \lambda_H\boldsymbol{n}$, where $\boldsymbol{n} = (\boldsymbol{N}_F, \boldsymbol{N}_M)$ and $\lambda_H$ are the eigenvectors and the eigenvalues respectively. In terms of $\boldsymbol{N}_F$ and $\boldsymbol{N}_M$, this is equivalent to the following matrix equations:

$$\begin{cases} \tilde{\boldsymbol{M}}^{-1}\boldsymbol{N}_F\boldsymbol{\Sigma_x}(\boldsymbol{I}_n - \tilde{\boldsymbol{F}}^T\tilde{\boldsymbol{F}}) - \boldsymbol{N}_F - \tilde{\boldsymbol{F}}\boldsymbol{N}_F^T\tilde{\boldsymbol{F}} = \lambda_H\boldsymbol{N}_F \\ \boldsymbol{N}_F\boldsymbol{\Sigma_x}\tilde{\boldsymbol{F}}^T + \tilde{\boldsymbol{F}}\boldsymbol{\Sigma_x}\boldsymbol{N}_F^T - \boldsymbol{N}_M = \lambda_H\boldsymbol{N}_M \end{cases} \tag{38}$$

It is straightforward to show that $\boldsymbol{H}$ is diagonalizable with eigenvectors in four subspaces represented by $\boldsymbol{n}_k$, $\boldsymbol{n}_m$, $\boldsymbol{n}_s$ and $\boldsymbol{t}$:

$$\boldsymbol{n}_k = (\boldsymbol{K}^{\mu,\nu}\tilde{\boldsymbol{F}}_\perp, 0), \quad \lambda_H = 1 - \frac{\lambda_{m+\nu}}{\lambda_\mu}, \quad dim(\boldsymbol{n}_k) = m(n-m) \tag{39}$$

$$\boldsymbol{n}_m = (0, \mathcal{M}), \quad \lambda_H = 1, \quad dim(\boldsymbol{n}_m) = m^2$$

$$\boldsymbol{n}_s = (\boldsymbol{S}^{i,j}\tilde{\boldsymbol{F}}, -(\lambda_i + \lambda_j)\boldsymbol{S}^{i,j}), \ i \geq j, \quad \lambda_H = 2, \quad dim(\boldsymbol{n}_s) = \frac{m(m+1)}{2}$$

$$\boldsymbol{t} = (\boldsymbol{\Omega}^{r,s}\tilde{\boldsymbol{F}}, (\lambda_s - \lambda_r)\boldsymbol{S}^{r,s}), \ r > s, \quad \lambda_H = 0, \quad dim(\boldsymbol{t}) = \frac{m(m-1)}{2}$$

In the above, $\boldsymbol{S}^{i,j}, \boldsymbol{\Omega}^{i,j} \in \mathbb{R}^{m \times m}$ are symmetric and skew-symmetric matrices respectively, $\mathcal{M} \in \mathbb{R}^{m \times m}$, and $\boldsymbol{K}^{\mu,\nu} \in \mathbb{R}^{m \times (n-m)}$ are two arbitrary matrices, and $\tilde{\boldsymbol{F}}_\perp \in \mathbb{R}^{(n-m) \times n}$ is a full-rank matrix whose row space is orthogonal to that of $\tilde{\boldsymbol{F}}$. ($\mu, i, j, r, s \in [m]$ and $\nu \in [n-m]$). First, we observe that all eigenvalues are real and non-negative. Additionally, it is easy to verify that the above eigenvectors do not form an orthogonal basis for the space [4]. This implies that the Jacobian of the average dynamics is not symmetric, and therefore the average dynamics do not follow a pure gradient flow (i.e. they are not curl-free).

### C.1.1 Non-orthogonal projections

The non-orthogonality of the eigenbasis for $\boldsymbol{H}$ is a main difference between the Similarity Matching network and other networks studied in the paper. This complicates the dynamics as the evolution of perturbations along tangent and normal subspaces may not be independent. However, as we will see below, we can use non-orthogonal projections to decouple the dynamics.

Let $\boldsymbol{N} \in \mathbb{R}^{K \times K}$ contain the eigenvectors of $\boldsymbol{H}$ as columns, where $K = mn + m^2$ is the dimension of the parameter space. In this space, let $\boldsymbol{n} \in \mathbb{R}^K$ be an arbitrary deviation from the solution manifold. Since $\boldsymbol{N}$ is full-rank, we can have the following decomposition:

$$\boldsymbol{n} = \boldsymbol{N}\boldsymbol{\pi}, \tag{40}$$

where $\boldsymbol{\pi} \in \mathbb{R}^K$ contains the coefficients of non-orthogonal projections. From the eigenvalue equation for $\boldsymbol{H}$, the average update (i.e. averaged over all data) for $\boldsymbol{n}$, becomes:

$$\Delta \boldsymbol{n} = -\boldsymbol{H}\boldsymbol{n} = -\boldsymbol{H}\boldsymbol{N}\boldsymbol{\pi} = -\boldsymbol{N}\boldsymbol{\Lambda}\boldsymbol{\pi} \tag{41}$$

where $\boldsymbol{\Lambda}$ has $\lambda_H$'s as diagonal entries, and in the rightmost equality we used $\boldsymbol{H}\boldsymbol{N} = \boldsymbol{N}\boldsymbol{\Lambda}$. Additionally, by differentiating Eq. 40, we can also represent the update as $\Delta \boldsymbol{n} = \boldsymbol{N}\Delta\boldsymbol{\pi}$. Equating this expression with Eq. 41, and multiplying from the left side by $(\boldsymbol{N}^T\boldsymbol{N})^{-1}\boldsymbol{N}^T$, we obtain:

$$\Delta \boldsymbol{\pi} = -\boldsymbol{\Lambda}\boldsymbol{\pi}. \tag{42}$$

Since $\boldsymbol{\Lambda}$ is diagonal, the dynamics for $\pi_i$ are decoupled. This motivates using the non-orthogonal projection for studying the dynamics. The above shows, for example, that coordinates $\pi_i$ corresponding to $\lambda_H = 0$ (tangent space) have purely diffusive dynamics with no flows. Similarly, non-negative $\lambda_H$ suggests mean-reverting dynamics for other coordinates.

Since $\boldsymbol{N}$ is known in our problem (see Eq. 39), we can, in principle, find projections coefficients $\boldsymbol{\pi}$ for arbitrary $\boldsymbol{n}$ by multiplying Eq. 40 from the left side by $(\boldsymbol{N}^T\boldsymbol{N})^{-1}\boldsymbol{N}^T$ to get: $\boldsymbol{\pi} = (\boldsymbol{N}^T\boldsymbol{N})^{-1}\boldsymbol{N}^T\boldsymbol{n}$. However, since each vector in this equation is a vectorized version of a pair of weight matrices $(\boldsymbol{F}, \boldsymbol{M})$, this could lead to tedious calculations in terms of the matrices. Instead, we use heuristics to find $\boldsymbol{\pi}$ for our problem. The right hand side of Eq. 40 can be restructured into:

$$\boldsymbol{n} = \boldsymbol{N}_k\boldsymbol{\pi}_k + \boldsymbol{N}_m\boldsymbol{\pi}_m + \boldsymbol{N}_s\boldsymbol{\pi}_s + \boldsymbol{T}\boldsymbol{\pi}_t. \tag{43}$$

where $\boldsymbol{N}_k \in \mathbb{R}^{K \times dim(\boldsymbol{n}_k)}$ has eigenvectors associated with subspace $\boldsymbol{n}_k$ as columns, $\boldsymbol{\pi}_k \in \mathbb{R}^{dim(\boldsymbol{n}_k)}$ has the corresponding projection coefficients, and similarly for other subspaces. The coefficients vectors $\boldsymbol{\pi}_k$, $\boldsymbol{\pi}_m$, $\boldsymbol{\pi}_s$ and $\boldsymbol{\pi}_t$, could further be indexed according to the defined coordinates within

---

[4]This can be verified by the definition of the Euclidean inner product between two vectors, $\boldsymbol{a} = (\boldsymbol{A}_F, \boldsymbol{A}_M)$ and $\boldsymbol{b} = (\boldsymbol{B}_F, \boldsymbol{B}_M)$ in the parameter space, which can be written in terms of $(\boldsymbol{F}, \boldsymbol{M})$ pair as: $\langle \boldsymbol{a}, \boldsymbol{b} \rangle = \mathrm{tr}(\boldsymbol{A}_F^T\boldsymbol{B}_F) + \mathrm{tr}(\boldsymbol{A}_M^T\boldsymbol{B}_M)$.

each subspace, e.g. $\pi_k^{\mu,\nu}$, $\pi_t^{r,s}$, etc. One can also write the above in terms of the weight matrices, which leads to two matrix equations associated with $F$ and $M$, respectively. If $n = (N_F, N_M)$, the equation corresponding to $N_F$ becomes:

$$N_F = \sum_{\mu,\nu} \pi_k^{\mu,\nu} K^{\mu,\nu} \tilde{F}_\perp + \sum_{i,j} \pi_s^{i,j} S^{i,j} \tilde{F} + \sum_{r,s} \pi_t^{r,s} \Omega^{r,s} \tilde{F}. \tag{44}$$

Interestingly, the above decomposition does not involve the $M$ component of the parameters, and is self-contained in terms of the $F$ component. Hence, the associated $\pi$ coefficients can be found from the standard matrix perturbation results. This leads to the following coefficients, which can be easily verified through simple algebra:

$$\pi_k^{\mu,\nu} = \mathrm{tr}\Big(K^{\mu,\nu} \tilde{F}_\perp N_F^T\Big), \quad \pi_s^{i,j} = \frac{1}{\sqrt{2(1+\delta_i^j)}} \mathrm{tr}\Big(S^{i,j} \tilde{F} N_F^T\Big), \quad \pi_t^{r,s} = \frac{1}{\sqrt{2}} \mathrm{tr}\Big(\Omega^{r,s} \tilde{F} N_F^T\Big)$$

$$\tag{45}$$

The first set of coefficients ($\pi_k^{\mu,\nu}$) are essentially the Euclidean projections on the subspace $n_k$. In retrospect, this makes sense as this subspace is orthogonal to other subspaces. In contrast, the equations for $\pi_s^{i,j}$ and $\pi_t^{r,s}$ deviate from orthogonal Euclidean projections.

## C.2 Fluctuations

We are now ready to calculate the fluctuation terms caused by online learning noise. The sample updates on the manifold can be derived by substituting $\tilde{M}$ and $\tilde{W}$ from Eq. 35 into Eq. 36, and performing change of variables to get:

$$g_*(x) = \begin{cases} \Delta F_*(x) = \Delta F(x)|_{\tilde{\theta}} = \eta \tilde{M}^{-1} \tilde{F} x x^T (I - \tilde{F}^T \tilde{F}) = \tilde{M}^{-1} \tilde{F} x_{||} x_\perp^T \\ \Delta M_*(x) = \Delta M(x)|_{\tilde{\theta}} = \eta(\tilde{F} x x^T \tilde{F}^T - \tilde{M}) \end{cases} \tag{46}$$

where we use the $*$ subscript to denote the quantities calculated on the manifold of solutions. As we saw above, the projection onto $n_k$ subspace could be calculated according to orthogonal Euclidean projection. This leads to:

$$\mathrm{proj}(g_*(x), n_K^{\mu,\nu}) = \mathrm{tr}\Big(\Delta F_*(x)^T K^{\mu,\nu} \tilde{F}_\perp\Big) = \mathrm{tr}\Big(x_\perp x_{||}^T \tilde{F}^T \tilde{M}^{-1} K^{\mu,\nu} \tilde{F}_\perp\Big) = \frac{x_\mu x_{m+\nu}}{\lambda_\mu}. \tag{47}$$

In the above, we used the fact that we can write $K^{\mu,\nu} \tilde{F}_\perp = q_\mu v_{m+\nu}^T$. The dynamics along $n_k$ is mean-reverting with eigenvalues $\lambda_H^{\mu,\nu} = 1 - \frac{\lambda_{m+\nu}}{\lambda_\mu}$. Substituting these into Eq. 20 results in the fluctuation covariance:

$$\langle (\rho^{\mu,\nu})^2 \rangle = \frac{\eta \langle x_\mu^2 x_{m+\nu}^2 \rangle}{2\lambda_\mu^2 \lambda_H^{\mu,\nu}} = \frac{\eta \lambda_{m+\nu}}{2\lambda_\mu(1 - \frac{\lambda_{m+\nu}}{\lambda_\mu})} \qquad \mu \in [1, m], \nu \in [1, n-m] \tag{48}$$

It is easy to show from Eq. 45 that the projection of $g_*$ onto $n_s$ and $t$ are zero, and onto $n_m$ is non-zero. However, the deviation along $n_m$ does not induce any tangential diffusion. This is because $\Delta F(x)|_{\tilde{\theta}+n_m} = M^{-1} \tilde{F} x_{||} x_\perp^T$ (this follows from calculating the sample update at point $(\tilde{F}, \tilde{M} + \mathcal{M})$ and performing algebraic simplification). Replacing this term in Eq. 45 to obtain the tangential projection leads to $\pi_t^{r,s} = 0$. Hence, we will skip calculating fluctuations in subspace $n_m$.

## C.3 Diffusion

Next, we proceed to calculate the diffusion into the tangent space. As we saw in the previous section, the relevant fluctuations are along $n_k$ subspace. Approximation of the sample update near the manifold shows:

$$\Delta M|_{\tilde{\theta}+\rho n_k}(x) \approx \eta(\tilde{F} x x^T \tilde{F}^T - \tilde{M} + \rho \tilde{F} x x^T N_F^T + \rho N_F x x^T \tilde{F}^T) \tag{49}$$

$$\Delta F|_{\tilde{\theta}+\rho n_k}(x) \approx \eta \tilde{M}^{-1}(\tilde{F} x x^T (I_n - \tilde{F}^T \tilde{F}) + \rho N_F x x^T (I_n - \tilde{F}^T \tilde{F}) - \rho \tilde{F} x x^T (\tilde{F}^T N_F + N_F^T \tilde{F}))$$

where $\boldsymbol{N}_F = \boldsymbol{K}^{\mu,\nu}\tilde{\boldsymbol{F}}_\perp$, and terms of $\mathcal{O}(\rho^2)$ and higher order are ignored. The non-orthogonal projection (which we denote by proj$^*$) onto the tangent space can be derived by calculating the corresponding $\pi_t$ from Eq. 45, to have:

$$\text{proj}^*(\boldsymbol{g}(x)|_{\tilde{\boldsymbol{\theta}}+\rho\boldsymbol{n}_k^{\mu,\nu}}, \boldsymbol{t}^{s,r}) = -\frac{1}{\sqrt{2}}\,\text{tr}\Big(\boldsymbol{\Omega}^{r,s}\Delta\boldsymbol{F}\tilde{\boldsymbol{F}}^T\Big) = \frac{\rho x_{m+\nu}}{\sqrt{2}}(\frac{x_s}{\lambda_s}\delta_\mu^r - \frac{x_r}{\lambda_r}\delta_\mu^s) \tag{50}$$

This form is very similar to Oja's case (see Eq. 32). Thus, the pairwise diffusion can be derived by summing up the average of the squared term above over all directions of the $\boldsymbol{n}_k$ and weighting each contribution by the corresponding fluctuation covariance in that direction. This yields:

$$D_{sr} = \frac{\eta^2}{8}\sum_{\mu\in[m]}\sum_{\nu\in[n-m]}\lambda_{m+\nu}(\frac{1}{\lambda_s}\delta_\mu^r + \frac{1}{\lambda_r}\delta_\mu^s)\langle(\rho^{\mu,\nu})^2\rangle \tag{51}$$

$$= \frac{\eta^3}{8}\sum_{\mu\in[m]}\sum_{\nu\in[n-m]}\lambda_{m+\nu}(\frac{1}{\lambda_s}\delta_\mu^r + \frac{1}{\lambda_r}\delta_\mu^s)\frac{\lambda_{m+\nu}}{2\lambda_\mu(1-\frac{\lambda_{m+\nu}}{\lambda_\mu})}$$

$$= \frac{\eta^3}{16\lambda_s\lambda_r}\sum_{\nu\in[n-m]}\lambda_{m+\nu}^2\Big(\frac{1}{1-\frac{\lambda_{m+\nu}}{\lambda_s}} + \frac{1}{1-\frac{\lambda_{m+\nu}}{\lambda_r}}\Big)$$

## D   Autoencoder

Here, we study a two-layer linear autoencoder with weights $\boldsymbol{U} \in \mathbb{R}^{p\times n}$ and $\boldsymbol{W} \in \mathbb{R}^{n\times p}$. The input is $\boldsymbol{x} \in \mathbb{R}^n$ and the network prediction at the output is $\hat{\boldsymbol{y}} = \boldsymbol{WU}\boldsymbol{x} \in \mathbb{R}^n$. We are interested in quantifying the drift of representation in the hidden layer $\boldsymbol{h} \in \mathbb{R}^p$. The learning occurs with SGD and batch size of one. The associated sample loss is:

$$l(\boldsymbol{x}, \boldsymbol{y}; \boldsymbol{\theta}) = \frac{1}{2}\|\boldsymbol{y}-\boldsymbol{WU}\boldsymbol{x}\|^2, \tag{52}$$

where the reconstruction requires $\boldsymbol{y} = \boldsymbol{x}$. We are interested in the case where the hidden layer is a bottleneck, i.e. $p < n$. In this case, the solutions satisfy:

$$\tilde{\boldsymbol{W}}\tilde{\boldsymbol{U}} = \begin{bmatrix} \boldsymbol{I}_p & 0 \\ 0 & 0 \end{bmatrix}_{n\times n} \tag{53}$$

where $\boldsymbol{I}_p \in \mathbb{R}^{p\times p}$ is an identity matrix. Let the SVD of the input covariance be $\boldsymbol{\Sigma_x} = \boldsymbol{VSQ}^T$. The *balanced solutions*, where both weights have the same scale, satisfy:

$$\tilde{\boldsymbol{U}} = \boldsymbol{QI}_{p,n}\boldsymbol{V}^T, \quad \tilde{\boldsymbol{W}} = \tilde{\boldsymbol{U}}^T. \tag{54}$$

Here, $\boldsymbol{I}_{p,n} \in \mathbb{R}^{p\times n}$ is a rectangular identity matrix with $[\boldsymbol{I}_{p,n}]_{i,j} = \delta_i^j$, and $\boldsymbol{Q} \in \mathbb{R}^{p\times p}$ and $\boldsymbol{V} \in \mathbb{R}^{n\times n}$ are two orthonormal matrices. Note that, at the solution, the $p$-principal subspace of data is represented in the hidden layer. Hence, the task-irrelevant subspace here is $(n - p)$-dimensional.

### D.1   Hessian

The updates follow stochastic gradient descent i.e. $\Delta\boldsymbol{\theta} = -\eta\boldsymbol{g}(\boldsymbol{x}; \boldsymbol{\theta}) \equiv (\boldsymbol{G_U}, \boldsymbol{G_W})$, where $\boldsymbol{G_U}$ and $\boldsymbol{G_W}$ are gradient matrices and can be derived by analytical differentiation of the loss:

$$\begin{cases} \boldsymbol{G_W} = (\boldsymbol{WU} - \boldsymbol{I}_n)\boldsymbol{x}\boldsymbol{x}^T\boldsymbol{U}^T \\ \boldsymbol{G_U} = \boldsymbol{W}^T(\boldsymbol{WU} - \boldsymbol{I}_n)\boldsymbol{x}\boldsymbol{x}^T \end{cases} \tag{55}$$

By approximating the average gradient near a solution point $(\tilde{\boldsymbol{W}}, \tilde{\boldsymbol{U}})$, we can form a set of Hessian equations $\boldsymbol{Hn} = \lambda_H\boldsymbol{n}$, where $\boldsymbol{n} = \text{vec}(\boldsymbol{N_W}) + \text{vec}(\boldsymbol{N_U}) \equiv (\boldsymbol{N_W}, \boldsymbol{N_U})$, for weight matrices $\boldsymbol{N_W} \in \mathbb{R}^{n\times p}$ and $\boldsymbol{N_U} \in \mathbb{R}^{p\times n}$, and eigenvalues $\lambda_H$. This leads to the following matrix equations:

$$\begin{cases} (\tilde{\boldsymbol{W}}\tilde{\boldsymbol{U}} - \boldsymbol{I}_n)\boldsymbol{\Sigma_x}\boldsymbol{N_U}^T + (\boldsymbol{N_W}\tilde{\boldsymbol{U}} + \tilde{\boldsymbol{W}}\boldsymbol{N_U})\boldsymbol{\Sigma_x}\tilde{\boldsymbol{U}}^T = \lambda_H\boldsymbol{N_W} \\ \boldsymbol{N_W}^T(\tilde{\boldsymbol{W}}\tilde{\boldsymbol{U}} - \boldsymbol{I}_n)\boldsymbol{\Sigma_x} + \tilde{\boldsymbol{W}}^T(\boldsymbol{N_W}\tilde{\boldsymbol{U}} + \tilde{\boldsymbol{W}}\boldsymbol{N_U})\boldsymbol{\Sigma_x} = \lambda_H\boldsymbol{N_U} \end{cases} \tag{56}$$

Similar to the previous networks, we can write arbitrary deviation from the solution manifold as the following.

$$\delta\boldsymbol{\theta} = \boldsymbol{n}_1 + \boldsymbol{n}_2 + \boldsymbol{n}_3 + \boldsymbol{t} \tag{57}$$
$$\boldsymbol{n}_1 = (\alpha\tilde{\boldsymbol{U}}_\perp^T(\boldsymbol{K}^{\mu,\nu})^T, \boldsymbol{K}^{\mu,\nu}\tilde{\boldsymbol{U}}_\perp), \quad \boldsymbol{n}_2 = (\boldsymbol{Z}^{i,j}\tilde{\boldsymbol{W}}, \tilde{\boldsymbol{W}}^T\boldsymbol{Z}^{i,j}),$$
$$\boldsymbol{n}_3 = (\boldsymbol{S}^{i,j}\tilde{\boldsymbol{W}}, -\tilde{\boldsymbol{W}}^T\boldsymbol{S}^{i,j}), \quad \boldsymbol{t} = (-\tilde{\boldsymbol{W}}\boldsymbol{\Omega}^{r,s}, \boldsymbol{\Omega}^{r,s}\tilde{\boldsymbol{W}}^T)$$

Importantly, the Hessian becomes diagonal in the above orthogonal coordinates. The detailed coordinates for each subspace and the associated Hessian eigenvalues are mentioned below. It is straightforward algebra to verify that each solution satisfies Eq. 56.

**Subspace $\boldsymbol{n}_1$:**

$$\boldsymbol{N}_{\boldsymbol{W}} = \alpha\boldsymbol{N}_{\boldsymbol{U}}^T, \quad \boldsymbol{N}_{\boldsymbol{U}} = \boldsymbol{K}^{\mu,\nu}\tilde{\boldsymbol{U}}_\perp = \boldsymbol{q}_\mu\boldsymbol{v}_{p+\nu}^T, \quad \text{where } [\boldsymbol{K}^{\mu,\nu}]_{ij} = q_{i\mu}\delta_j^\nu, \quad \mu \in [p], \nu \in [n-p],$$
$$\lambda_H^{\mu,\nu} = \lambda_{p+\nu}(1 - \alpha_{\mu,\nu}), \quad dim(\boldsymbol{n}_1) = 2p(n-p)$$

In the above, $\boldsymbol{K}^{\mu,\nu} \in \mathbb{R}^{p\times(n-p)}$, and $\tilde{\boldsymbol{U}}_\perp \in \mathbb{R}^{(n-p)\times n}$ is a full-rank matrix whose row space is orthogonal to that of $\tilde{\boldsymbol{U}}$. Also, recall that $\boldsymbol{q}_i$ and $\boldsymbol{v}_i$ are the columns of orthonormal matrices in Eq. 54. Replacing the above in the Hessian equations, yields in the following characteristic quadratic equation for $\alpha_{\mu,\nu}$ :

$$\lambda_{p+\nu}\alpha_{\mu,\nu}^2 + (\lambda_\mu - \lambda_{p+\nu})\alpha_{\mu,\nu} - \lambda_{p+\nu} = 0, \tag{58}$$
$$\text{with solutions:} \quad \alpha_{\mu,\nu} = \frac{1}{2\lambda_{p+\nu}}\left(-(\lambda_\mu - \lambda_{p+\nu}) \pm \sqrt{(\lambda_\mu - \lambda_{p+\nu})^2 + 4\lambda_{p+\nu}^2}\right).$$

Note that normally there are two solutions to the above, which leads to two different eigenvectors for each pair $(\mu, \nu)$.

**Subspace $\boldsymbol{n}_2$:**

$$\boldsymbol{N}_{\boldsymbol{W}} = \boldsymbol{Z}^{i,j}\tilde{\boldsymbol{W}}, \quad \boldsymbol{N}_{\boldsymbol{U}} = \tilde{\boldsymbol{W}}^T\boldsymbol{Z}^{i,j}, \quad \text{where } \boldsymbol{Z}^{i,j} = \boldsymbol{v}_i\boldsymbol{v}_j^T, \quad i, j \in [p]$$
$$\lambda_H^{i,j} = 2\lambda_i, \quad dim(\boldsymbol{n}_2) = p^2.$$

**Subspace $\boldsymbol{n}_3$:**

$$\boldsymbol{N}_{\boldsymbol{W}} = \boldsymbol{S}^{i,j}\tilde{\boldsymbol{W}}, \quad \boldsymbol{N}_{\boldsymbol{U}} = -\boldsymbol{N}_{\boldsymbol{W}}^T, \quad \text{where } \boldsymbol{S}^{i,j} = \boldsymbol{v}_i\boldsymbol{v}_j^T + \boldsymbol{v}_j\boldsymbol{v}_i^T, \quad i, j \in [p]$$
$$\lambda_H^{i,j} = 0, \quad dim(\boldsymbol{n}_3) = \frac{p(p+1)}{2}$$

**Subspace $\boldsymbol{t}$:** (tangent space)

$$\boldsymbol{T}_{\boldsymbol{W}} = \boldsymbol{T}_{\boldsymbol{U}}^T, \quad \boldsymbol{T}_{\boldsymbol{U}} = \boldsymbol{\Omega}^{r,s}\tilde{\boldsymbol{U}}, \quad \text{where } \boldsymbol{\Omega}^{r,s} = \frac{1}{2}(\boldsymbol{q}_r\boldsymbol{q}_s^T - \boldsymbol{q}_s\boldsymbol{q}_r^T), \quad r, s \in [p]$$
$$\lambda_H^{r,s} = 0 \quad dim(\boldsymbol{t}) = \frac{p(p-1)}{2}$$

In the above, subspace $\boldsymbol{n}_3$ corresponds to deviation from the balanced solution; movements along this subspace scale the weights $\boldsymbol{W}$ and $\boldsymbol{U}$ inversely, such that their product remains fixed. Hence, this is technically also a tangent space. As we are interested in the drift associated with rotational symmetry, we only calculate the drift in the tangential subspace $\boldsymbol{t}$.

### D.2 Fluctuation

By replacing $(\tilde{\boldsymbol{W}}, \tilde{\boldsymbol{U}})$ into Eq. 55, the sample gradient on the solution manifold becomes:

$$\boldsymbol{g}_*(\boldsymbol{x}; \tilde{\boldsymbol{\theta}}) = \begin{cases} \boldsymbol{G}_{\boldsymbol{W}} = -\boldsymbol{I}_\perp \boldsymbol{x}\boldsymbol{x}^T\tilde{\boldsymbol{U}}^T \\ \boldsymbol{G}_{\boldsymbol{U}} = 0 \end{cases} \tag{59}$$

($\boldsymbol{I}_\perp$ is the projection operator onto the task-irrelevant subspace). The inner product of two vectors $\boldsymbol{a}$ and $\boldsymbol{b}$ can be written in terms of traces of the corresponding weight matrices. This means $\langle\boldsymbol{a}, \boldsymbol{b}\rangle =$

$\mathrm{tr}\left(\boldsymbol{A}_W^T \boldsymbol{B}_W\right) + \mathrm{tr}\left(\boldsymbol{A}_U^T \boldsymbol{B}_U\right)$. Using this equation and some linear algebra, it is straightforward to show that $\boldsymbol{g}_*$ has a non-zero projection on $\boldsymbol{n}_1$, while projections on $\boldsymbol{n}_2$ and $\boldsymbol{n}_3$ and $\boldsymbol{t}$ are zero. After normalization, the projection onto $\boldsymbol{n}_1$ becomes:

$$\mathrm{proj}(\boldsymbol{n}_1^{\mu,\nu}, \boldsymbol{g}_*) = \frac{\alpha_{\mu,\nu} x_\mu x_{p+\nu}}{\sqrt{1 + \alpha_{\mu,\nu}^2}}, \tag{60}$$

where $\alpha_{\mu,\nu}$ are functions of $\lambda_\mu$ and $\lambda_{p+\nu}$ as was shown in Eq. 58. Replacing this into Eq. 20, we get the covariance of fluctuations in subspace $\boldsymbol{n}_1$:

$$\langle \rho_{\mu,\nu}^2 \rangle = \frac{\eta}{2\lambda_H^{\mu,\nu}} \frac{\alpha_{\mu,\nu}^2}{1 + \alpha_{\mu,\nu}^2} \langle x_\mu^2 x_{p+\nu}^2 \rangle = \frac{\eta \alpha_{\mu,\nu}^2 \lambda_\mu}{2(1 + \alpha_{\mu,\nu}^2)(1 - \alpha_{\mu,\nu})}. \tag{61}$$

This is the fluctuation of the task-irrelevant representations toward task-relevant ones.

### D.3  Diffusion

To find the diffusion induced by the fluctuations in subspace $\boldsymbol{n}_1$, we have to approximate the gradient in Eq. 55 at a point $\tilde{\boldsymbol{\theta}} + \rho \boldsymbol{n}_1^{\mu,\nu}$ near the manifold and project that onto the tangent space ($\boldsymbol{t}$). After some algebra, the projection to the leading order in $\rho$ becomes:

$$\mathrm{proj}(\boldsymbol{g}(\boldsymbol{x}; \tilde{\boldsymbol{\theta}} + \rho \boldsymbol{n}_1^{\mu,\nu}), \boldsymbol{t}^{s,r}) = \frac{1}{2} \rho x_{p+\nu}(x_r \delta_\mu^s - x_s \delta_\mu^r) \tag{62}$$

In reference to Eq. 22, the $\mathcal{G}_{\mu,\nu}^{s,r}(x)$ coefficients become:

$$\mathcal{G}_{\mu,\nu}^{s,r} = \frac{1}{4} x_{p+\nu}(x_r \delta_\mu^s - x_s \delta_\mu^r). \tag{63}$$

Subsequently, the the pairwise diffusion rates $D_{sr}$, for $r, s \in [p]$, $r > s$ becomes:

$$D_{sr} = \frac{\eta^2}{2} \sum_{\mu \in [p]} \sum_{\nu \in [n-p]} \langle \rho_{\mu\nu}^2 \rangle \langle (\mathcal{G}_{\mu,\nu}^{s,r})^2 \rangle_x \tag{64}$$

$$= \frac{\eta^2}{2} \sum_{i \in \{1,2\}} \sum_{\mu \in [p]} \sum_{\nu \in [n-p]} [\frac{\eta \alpha_{\mu,\nu,i}^2 \lambda_\mu}{2(1 + \alpha_{\mu,\nu,i}^2)(1 - \alpha_{\mu,\nu,i})}][\frac{1}{16} \lambda_{p+\nu}(\lambda_r \delta_\mu^s + \lambda_s \delta_\mu^r)]$$

$$= \frac{\eta^3}{64} \sum_{i \in \{1,2\}} \sum_{\nu \in [n-p]} \lambda_{p+\nu} \lambda_r \lambda_s [\frac{\alpha_{s,\nu,i}^2}{(1 + \alpha_{s,\nu,i}^2)(1 - \alpha_{s,\nu,i})} + \frac{\alpha_{r,\nu,i}^2}{(1 + \alpha_{r,\nu,i}^2)(1 - \alpha_{r,\nu,i})}]$$

($\alpha_{\mu,\nu,i}$, $i \in \{1,2\}$ are the two solutions for $\alpha$ — see Eq. 58). The above could be written in a more compact form:

$$D_{sr} = \frac{\eta^3 \lambda_r \lambda_s}{64} \sum_{\nu \in [n-p]} \lambda_{p+\nu}(f(\lambda_s, \lambda_{p+\nu}) + f(\lambda_r, \lambda_{p+\nu}))$$

where $f(\lambda_\mu, \lambda_{p+\nu}) = \sum_{\alpha: Q(\alpha)=0} \frac{\alpha^2}{(1+\alpha^2)(1-\alpha)}$, and $Q(\alpha) = \lambda_{p+\nu}\alpha^2 + (\lambda_\mu - \lambda_{p+\nu})\alpha - \lambda_{p+\nu}$.

## E  Two-Layer Network (Supervised)

Here, we study a linear two-layer network trained with supervised learning. The network output prediction is $\hat{\boldsymbol{y}} = \boldsymbol{WUx}$, where $\boldsymbol{W} \in \mathbb{R}^{m \times p}$, and $\boldsymbol{U} \in \mathbb{R}^{p \times n}$ are two weight matrices ($m, n \leqslant p$). The network is trained with a teacher that dictates the relationship $\boldsymbol{y} = \boldsymbol{Px}$ between the input and the output ($\boldsymbol{P} \in \mathbb{R}^{m \times n}$). We are interested in the drift of representations in the hidden layer $\boldsymbol{h} \in \mathbb{R}^p$ at the steady state. Specifically, we would like to study how the stimuli that are task-relevant and irrelevant contribute to drift. Note that here, the task-irrelevant stimuli are the ones that lie in the null-space of $\boldsymbol{P}$, and hence this is a more general case than the principal subspace studied for other networks. Additionally, we allow for the possibility that the mapping $\boldsymbol{P}$ is low-rank, and denote its rank with $k \triangleq rank(\boldsymbol{P}) \leqslant m$. This implies that the task-irrelevant data lie in the $(n - k)$-dimensional null-space of $\boldsymbol{P}$. For simplicity, we assume that the non-zero singular values of $\boldsymbol{P}$ are equal to one. This allows us to express it as $\boldsymbol{P} = \boldsymbol{R} \boldsymbol{I}_{m,n}^{k} \bar{\boldsymbol{V}}^T$, where $\boldsymbol{R} \in \mathbb{R}^{m \times m}$ and $\bar{\boldsymbol{V}} \in \mathbb{R}^{n \times n}$ are two arbitrary orthonormal matrices, and $\boldsymbol{I}_{m,n}^{k} \triangleq \boldsymbol{I}_{m,k} \boldsymbol{I}_{k,n}$ denotes a rectangular identity matrix whose first $k$ diagonal entries are one, and all others are zero. Finally, note that in [25], drift in a two-layer autoencoder with an expansive layer was studied. Our setup is similar to that work but is more general, as the task is not limited to reconstructing the input.

The sample gradient can be obtained by differentiating the MSE loss for the two-layer network:

$$\begin{cases} \boldsymbol{G_W} = (\boldsymbol{WU} - \boldsymbol{P})\boldsymbol{xx}^T\boldsymbol{U}^T + \gamma \boldsymbol{W} \\ \boldsymbol{G_U} = \boldsymbol{W}^T(\boldsymbol{WU} - \boldsymbol{P})\boldsymbol{xx}^T + \gamma \boldsymbol{U} \end{cases} \tag{65}$$

where $\gamma$ is the weight-decay coefficient. Correspondingly, the average gradient is:

$$\begin{cases} \langle \boldsymbol{G_W} \rangle = (\boldsymbol{WU} - \boldsymbol{P})\boldsymbol{\Sigma_x}\boldsymbol{U}^T + \gamma \boldsymbol{W} \\ \langle \boldsymbol{G_U} \rangle = \boldsymbol{W}^T(\boldsymbol{WU} - \boldsymbol{P})\boldsymbol{\Sigma_x} + \gamma \boldsymbol{U} \end{cases} \tag{66}$$

where the averages $\langle . \rangle$ are taken with respect to the data. We are first interested in finding the manifold of solution, i.e. set of $(\tilde{\boldsymbol{W}}, \tilde{\boldsymbol{U}})$ that satisfy $\langle \boldsymbol{G_W} \rangle = 0$ and $\langle \boldsymbol{G_U} \rangle = 0$ simultaneously. To do so, we first perform the following change of variables to "primed" weight matrices:

$$\boldsymbol{W} = \boldsymbol{R} \boldsymbol{I}_{m,k} \boldsymbol{W}', \quad \boldsymbol{U} = \boldsymbol{U}' \boldsymbol{I}_{k,n} \bar{\boldsymbol{V}}^T, \tag{67}$$

where $\boldsymbol{W}' \in \mathbb{R}^{k \times p}$ and $\boldsymbol{U}' \in \mathbb{R}^{p \times k}$, and can be considered as weights of a reduced two-layer network (see below). Replacing the above in Eq. 66, leads to the corresponding gradient equations for the primed variables:

$$\begin{cases} \langle \boldsymbol{G}'_{\boldsymbol{W}} \rangle = (\boldsymbol{W}'\boldsymbol{U}' - \boldsymbol{I}_k)\boldsymbol{\Sigma}'_{\boldsymbol{x}}\boldsymbol{U}'^T + \gamma \boldsymbol{W}' \\ \langle \boldsymbol{G}'_{\boldsymbol{U}} \rangle = \boldsymbol{W}'^T(\boldsymbol{W}'\boldsymbol{U}' - \boldsymbol{I}_k)\boldsymbol{\Sigma}'_{\boldsymbol{x}} + \gamma \boldsymbol{U}' \end{cases} \tag{68}$$

Here, $\boldsymbol{I}_k \in \mathbb{R}^{k \times k}$ is an identity matrix, and $\boldsymbol{\Sigma}'_{\boldsymbol{x}} = \boldsymbol{I}_{k,n} \bar{\boldsymbol{V}}^T \boldsymbol{\Sigma_x} \bar{\boldsymbol{V}} \boldsymbol{I}_{n,k}$ is the effective covariance in the primed coordinates. These equations correspond to a reduced network, which is a two-layer expansive autoencoder with input covariance $\boldsymbol{\Sigma}'_{\boldsymbol{x}} \in \mathbb{R}^{k \times k}$ (this sub-network can be interpreted as dealing with the task-relevant portion of the data). The manifold of solutions for this two-layer expansive autoencoder has been previously derived in [25], and satisfies: $\tilde{\boldsymbol{W}}'\tilde{\boldsymbol{W}}'^T = \boldsymbol{I}_k - \gamma \boldsymbol{\Sigma}'^{-1}_{\boldsymbol{x}}$ and $\tilde{\boldsymbol{U}}' = \tilde{\boldsymbol{W}}'^T$. From these, the solutions to the non-primed variables can be derived by the change of variables of Eq. 67.

We will solve drift for Gaussian data $\boldsymbol{x} \sim \mathcal{N}(0, \boldsymbol{\Sigma_x})$. Similar to other networks, we assume the eigenvalues of the input covariance in the task-relevant and task-irrelevant subspaces are $\lambda_{||} = 1$ and $\lambda_\perp$, respectively. However, note that here the task is determined by $\boldsymbol{P}$, and the multiplicity of $\lambda_{||}$ and $\lambda_\perp$ are $k$ and $n - k$, respectively. Additionally, and unlike in previous cases of the principal subspace task, $\lambda_\perp$ is not restricted to be smaller than $\lambda_{||}$. This data structure simplifies the effective covariance in Eq. 68 to be $\boldsymbol{\Sigma}'_{\boldsymbol{x}} = \boldsymbol{I}_k$. This leads to the weight solutions in the primed coordinates to be $\tilde{\boldsymbol{W}}' = \sqrt{1 - \gamma}\boldsymbol{I}_{k,p}\boldsymbol{Q}_p^T$ and $\tilde{\boldsymbol{U}}' = \sqrt{1 - \gamma}\boldsymbol{Q}_p\boldsymbol{I}_{p,k}$, where $\boldsymbol{Q}_p \in \mathbb{R}^{p \times p}$ is an arbitrary orthonormal matrix. Finally, by the change of variables in Eq. 67, the manifold of solutions for the original network becomes:

$$\tilde{\boldsymbol{W}} = \sqrt{1 - \gamma}\boldsymbol{R} \boldsymbol{I}_{m,k}\boldsymbol{I}_{k,p}\boldsymbol{Q}_p^T, \quad \tilde{\boldsymbol{U}} = \sqrt{1 - \gamma}\boldsymbol{Q}_p\boldsymbol{I}_{p,k}\boldsymbol{I}_{k,n}\bar{\boldsymbol{V}}^T. \tag{69}$$

## E.1 Hessian

By expanding the average gradient (Eq. 66) near a solution point $(\tilde{W}, \tilde{U})$, and similar to the autoencoder case, we can form a set of Hessian equations $Hn = \lambda_H n$, where $n = \mathrm{vec}(N_W) + \mathrm{vec}(N_U) \equiv (N_W, N_U)$, for weight matrices $N_W \in \mathbb{R}^{m \times p}$ and $N_U \in \mathbb{R}^{p \times n}$.

$$-\gamma P N_U^T + (N_W \tilde{U} + \tilde{W} N_U) \Sigma_x \tilde{U}^T + \gamma N_W = \lambda_H N_W \qquad (70)$$
$$-\gamma N_W^T P + \tilde{W}^T (N_W \tilde{U} + \tilde{W} N_U) \Sigma_x + \gamma N_U = \lambda_H N_U$$

One can show that the Hessian is diagonalized in the coordinates discussed below.

**Subspace $n_1$:**
$$N_W = 0, \quad N_U^{\mu,\nu} = K^{\mu,\nu} P_\perp = q_\mu \bar{v}_{k+\nu}^T, \quad \text{where } [K^{\mu,\nu}]_{ij} = q_{i\mu} \delta_j^\nu \quad \mu \in [k], \ \nu \in [n-k],$$
$$\lambda_H^{\mu,\nu} = \lambda_\perp (1 - \gamma) + \gamma, \quad dim(n_1) = k(n-k)$$

In the above $K^{\mu,\nu} \in \mathbb{R}^{p \times (n-k)}$ and $P_\perp \in \mathbb{R}^{(n-k) \times n}$ is a full-rank matrix whose row space is orthogonal to that of $P$. Also note that in the above and the rest of the section, $q_i$ and $\bar{v}_i$ are columns of matrices $Q$ and $\bar{V}$ respectively.

**Subspace $n_2$:**
$$N_W = 0, \quad N_U^{\mu,\nu} = K^{\mu,\nu} P_\perp = q_\mu \bar{v}_{k+\nu}^T, \quad \text{where } [K^{\mu,\nu}]_{ij} = q_{i\mu} \delta_j^\nu \quad \mu \in [k+1,p], \ \nu \in [n-k],$$
$$\lambda_H^{\mu,\nu} = \gamma, \quad dim(n_2) = (p-k)(n-k)$$

**Subspace $n_3$:**
$$N_W = R I_{m,k} N'_W, \quad N_U = -N_W'^T I_{k,n} \bar{V}^T, \quad \text{where } N'_W = S\tilde{W}' + K\tilde{W}'_\perp$$
$$\lambda_H = 2\gamma, \quad dim(n_3) = kp - \frac{k(k-1)}{2}.$$

In the above, $S \in \mathbb{R}^{k \times k}$ is a symmetric matrix, $K \in \mathbb{R}^{k \times (p-k)}$ is an arbitrary matrix, and $\tilde{W}'_\perp \in \mathbb{R}^{(p-k) \times p}$ is full-rank matrix whose row-space is orthogonal to that of $\tilde{W}'$.

**Subspace $n_4$:**
$$N_W = R I_{m,k} Z^{i,j} \tilde{W}', \quad N_U = \tilde{W}'^T Z^{i,j} I_{k,n} \bar{V}^T, \quad Z^{i,j} \in \mathbb{R}^{k \times k}, \quad i,j \in [k] \quad dim(n_4) = k^2$$

where $Z^{i,j}$ consist of three subgroups:

$$Z^{i,i} = \frac{1}{\sqrt{2(1-\gamma)}} z_i z_i^T, \quad Z_{(i>j)}^{i,j} = \frac{1}{2\sqrt{1-\gamma}} (z_i z_j^T + z_j z_i^T), \quad \lambda_H = 2(1-\gamma)$$

$$Z_{(i>j)}^{i,j} = \frac{1}{2\sqrt{1-\gamma}} (z_i z_j^T - z_j z_i^T), \quad \lambda_H = 2$$

and set of $z_i \in \mathbb{R}^k$ form an orthonormal basis for $\mathbb{R}^k$.

**Subspace $n_5$:**
$$N_W^{\mu,\nu} = r_{k+\nu} q_\mu^T, \quad N_U = 0, \quad \text{where } \mu \in [k], \ \nu \in [m-k], \quad \lambda_H^{\mu,\nu} = 1, \quad dim(n_5) = k(m-k)$$
(in the above, $r_i$'s are the columns of the orthonormal matrix $R$).

**Subspace $n_6$:**
$$N_W^{\mu,\nu} = r_{k+\nu} q_\mu^T, \quad N_U = 0, \quad \text{where } \mu \in [k+1,p], \ \nu \in [m-k],$$
$$\lambda_H^{\mu,\nu} = \gamma, \quad dim(n_6) = (p-k)(m-k)$$

**Subspace $t_1$:** (tangent space)

$$T_W = \tilde{W} \Omega^{r,s}, \quad T_U = -\Omega^{r,s} \tilde{U}, \quad \text{where } \Omega^{r,s} = \frac{1}{2\sqrt{1-\gamma}} (q_r q_s^T - q_s q_r^T), \quad r \neq s, \ r,s \in [k]$$

$$\lambda_H^{r,s} = 0 \qquad dim(t_1) = \frac{k(k-1)}{2}$$

**Subspace $t_2$:**  (tangent space)

$$\boldsymbol{T}_W = \boldsymbol{r}_\mu \boldsymbol{q}_{k+\nu}^T, \quad \boldsymbol{T}_U = \boldsymbol{q}_{k+\nu} \bar{\boldsymbol{v}}_\mu^T, \quad \text{where } \mu \in [k], \nu \in [p-k]$$

$$\lambda_H^{\mu,\nu} = 0 \qquad dim(\boldsymbol{t}_2) = k(p-k)$$

The first tangent space ($\boldsymbol{t}_1$) corresponds to rotation of representations within the row-space of $\tilde{\boldsymbol{W}}$, while the second one ($\boldsymbol{t}_2$) corresponds to rotation toward the $(p-k)$-dimensional subspace orthogonal to the row-space of $\tilde{\boldsymbol{W}}$.

### E.2   Fluctuations

Sample gradient on the manifold of solution can be calculated from Eq. 65:

$$\boldsymbol{g}_*(\boldsymbol{x}; \tilde{\boldsymbol{\theta}}) = \begin{cases} \boldsymbol{G}_W = \gamma(\tilde{\boldsymbol{W}} - \boldsymbol{P}\boldsymbol{x}\boldsymbol{x}^T \tilde{\boldsymbol{U}}^T) \\ \boldsymbol{G}_U = \gamma(\tilde{\boldsymbol{U}} - \tilde{\boldsymbol{W}}^T \boldsymbol{P}\boldsymbol{x}\boldsymbol{x}^T) \end{cases} \tag{71}$$

The projection on subspace $\boldsymbol{n}_1$ is:

$$\text{proj}(\boldsymbol{g}_*, \boldsymbol{n}_1^{\mu,\nu}) = -\gamma\sqrt{1-\gamma} x_\mu x_{k+\nu}, \tag{72}$$

where the indices here are with respect to the columns of matrix $\bar{\boldsymbol{V}}$, i.e. $x_\mu := \bar{\boldsymbol{v}}_\mu^T \boldsymbol{x}$, etc. The above projection leads to the following covariance for the fluctuations (using Eq. 20):

$$\langle \rho_{\mu\nu}^2 \rangle = \frac{\eta\gamma^2(1-\gamma)}{2\lambda_H^{\mu,\nu}} \langle x_\mu^2 x_\nu^2 \rangle = \frac{\eta\gamma^2}{2} \frac{\lambda_\perp(1-\gamma)}{\lambda_\perp(1-\gamma)+\gamma}. \tag{73}$$

We can also show that there is a non-zero projection on subspace $\boldsymbol{n}_4$:

$$\text{proj}(\boldsymbol{g}_*, \boldsymbol{n}_4^{i,i}) = -\sqrt{2}\gamma\sqrt{1-\gamma}(1-x_i^2), \quad \text{proj}(\boldsymbol{g}_*, \boldsymbol{n}_4^{i,j}) = -\gamma\sqrt{1-\gamma}x_i x_j \ (i \neq j) \tag{74}$$

Projections onto other subspaces are zero.

### E.3   Diffusion

To find the diffusion induced by the fluctuations in subspace $\boldsymbol{n}_1$, we have to approximate the gradient in Eq. 65 at a point $\tilde{\boldsymbol{\theta}} + \rho\boldsymbol{n}_1^{\mu,\nu}$ near the manifold and project that onto the tangent spaces. Since for subspace $\boldsymbol{n}_1$, we had $\boldsymbol{n}_1 = (0, \boldsymbol{N}_U)$, the approximate gradient becomes:

$$\boldsymbol{G}_W|_{\tilde{\boldsymbol{\theta}}+\rho\boldsymbol{n}_1^{\mu,\nu}} = (\tilde{\boldsymbol{W}}(\tilde{\boldsymbol{U}} + \rho\boldsymbol{N}_U) - \boldsymbol{P})\boldsymbol{x}\boldsymbol{x}^T(\tilde{\boldsymbol{U}}^T + \rho\boldsymbol{N}_U^T) + \gamma\tilde{\boldsymbol{W}} \tag{75}$$

$$\approx \gamma(\tilde{\boldsymbol{W}} - \boldsymbol{P}\boldsymbol{x}\boldsymbol{x}^T\tilde{\boldsymbol{U}}^T) + \rho(\tilde{\boldsymbol{W}}\boldsymbol{N}_U\boldsymbol{x}\boldsymbol{x}^T\tilde{\boldsymbol{U}}^T - \gamma\boldsymbol{P}\boldsymbol{x}\boldsymbol{x}^T\boldsymbol{N}_U^T)$$

$$\boldsymbol{G}_U|_{\tilde{\boldsymbol{\theta}}+\rho\boldsymbol{n}_1^{\mu,\nu}} = \tilde{\boldsymbol{W}}^T(\tilde{\boldsymbol{W}}(\tilde{\boldsymbol{U}} + \rho\boldsymbol{N}_U) - \boldsymbol{P})\boldsymbol{x}\boldsymbol{x}^T + \gamma\tilde{\boldsymbol{U}} + \gamma\rho\boldsymbol{N}_U$$

$$\approx \gamma(\tilde{\boldsymbol{U}} - \tilde{\boldsymbol{W}}^T\boldsymbol{P}\boldsymbol{x}\boldsymbol{x}^T) + \rho(\tilde{\boldsymbol{W}}^T\tilde{\boldsymbol{W}}\boldsymbol{N}_U\boldsymbol{x}\boldsymbol{x}^T + \gamma\boldsymbol{N}_U).$$

In the above, we ignored terms of $\mathcal{O}(\rho^2)$ and higher orders. Replacing $\boldsymbol{N}_U^{\mu,\nu} = \boldsymbol{q}_\mu \bar{\boldsymbol{v}}_{k+\nu}^T$ in the above and performing the projection leads to:

$$\text{proj}(\boldsymbol{g}(\boldsymbol{x}; \tilde{\boldsymbol{\theta}} + \rho\boldsymbol{n}_1^{\mu,\nu}), \boldsymbol{t}^{s,r}) = \frac{1}{2}\gamma\rho x_{k+\nu}(x_s\delta_\mu^r - x_r\delta_\mu^s) \tag{76}$$

(Projection onto $\boldsymbol{t}_2$ is zero). Correspondingly, the pairwise diffusion coefficients become:

$$D_{sr}^{\boldsymbol{n}_1} = \frac{\eta^2}{2} \sum_{\mu \in [k]} \sum_{\nu \in [n-k]} \langle \rho_{\mu\nu}^2 \rangle \langle (\mathcal{G}_{\mu,\nu}^{s,r})^2 \rangle_x \qquad r, s \in [k], r \neq s \tag{77}$$

$$= \frac{\eta^2\gamma^2}{16(1-\gamma)} \sum_{\nu \in [n-k]} \lambda_{k+\nu} \langle \rho_{\mu\nu}^2 \rangle = \frac{\eta^3\gamma^4}{32} \frac{(n-k)\lambda_\perp^2}{\lambda_\perp(1-\gamma)+\gamma}$$

We also have a diffusion term that is caused by the fluctuations in the $\boldsymbol{n}_4$ subspace. This corresponds to diffusion of an expansive autoencoder with input dimension $k$ and an effective isotropic data, which has been shown previously to be [25]:

$$D_{sr}^{\boldsymbol{n}_4} = \frac{\eta^3\gamma^4(k+2)}{16(1-\gamma)} \tag{78}$$

Hence, total diffusion of the representation for stimulus $s$ becomes:

$$D_s = \frac{\eta^3 \gamma^4}{16(1-\gamma)}(k-1)(k+2+\frac{(n-k)}{2}\frac{\lambda_\perp^2(1-\gamma)}{\lambda_\perp(1-\gamma)+\gamma}) \tag{79}$$

$$\approx \frac{\eta^3 \gamma^4}{16}(k-1)(k+2+\frac{(n-k)\lambda_\perp}{2}), \quad \gamma \ll \lambda_\perp$$

This summarizes the derivation of drift for a two-layer supervised setup with teacher signal $\boldsymbol{y} = \boldsymbol{Px}$. Note that, unlike the principal subspace tracking networks, here $\lambda_\perp$ is the input variance in the null-space of $\boldsymbol{P}$, and is not restricted to be smaller than $\lambda_{||}$ (Figure S1a). Additionally, we see that in this case drift does not depend on the output dimension, but only on the input dimension and $k$, which is the rank of $\boldsymbol{P}$ (Figure S1b,c).

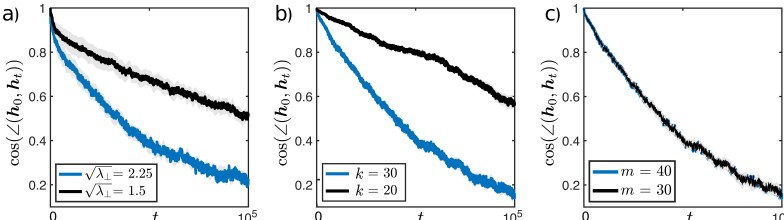

Figure S1: Simulations of drift in the supervised two-layer network. In all the plots, the cosine similarity decay of a task-relevant representation is shown over time. a) For a given $\boldsymbol{P}$, drift rate increases with the task-irrelevant variance $\lambda_\perp$. Entries of $\boldsymbol{P}$ are chosen randomly from a Gaussian distribution followed by a rescaling to set the maximum singular value to be one. The task-irrelevant subspace is the null-space of $\boldsymbol{P}$, and $\lambda_{||} = 1$. b,c) Drift rate changes with $k$ (rank of $\boldsymbol{P}$, panel b) and not $m$ (output dimension, panel c). Here, the singular values of $\boldsymbol{P}$ and the input covariance are set to one. Simulation parameters for each panel are a: $n = 50, p = 70, k = m = 30, \eta = 0.02, \gamma = 0.2$, b: $n = 50, p = 70, m = 40$, and c: $n = 50, p = 70, k = 30$.

