# OpenReview forum: "Contribution of task-irrelevant stimuli to drift of neural representations"
_NeurIPS.cc/2025/Conference — NeurIPS 2025 poster_

### Official Review · Reviewer_DyoC · 2025-07-01

**Clarity:** 3
**Significance:** 3
**Originality:** 3
**Rating:** 5
**Confidence:** 3

**Summary:**

In this paper, the authors show how the task-irrelevant portion of input can lead to a drift in internal representations of networks even after the learning in the relevant subspace has stabilized. The authors derive theoretical results for such drift for several network architectures and learning rules, establishing its dependence on factors like the learning rate and the dimension of the task-irrelevant subspace. Numerical results are presented which agree closely with the theory.

**Questions:**

See strengths and weaknesses

**Ethical Concerns:**

["NO or VERY MINOR ethics concerns only"]

**Final Justification:**

I want to thank the authors for addressing my minor concerns, and I maintain my favorable rating.

**Limitations:**

Yes

**Quality:**

3

**Strengths And Weaknesses:**

The paper is well-written, and the ideas are presented in a way making them easy to follow. The motivating example is quite useful for developing intuition before jumping into the theory. I was unfamiliar with a fair bit of the background work, but the paper does an excellent job at providing enough pointers to not have to jump back and forth between references in order to understand the core ideas. Interestingly, even though the results are developed for linear networks, they seem to generalize to non-linear ones leaving venues for future investigations. The results show close correspondence between theory and simulations. Overall, I recommend the paper to be accepted. Some concerns:

1. While the work provides insight into drift in representational learning, it likely has limited practical applications owing to the limitations authors acknowledge – simple linear networks, assumptions around small perturbations making a differential analysis possible.
2. Minor: in Fig. 1 consider using another variable name for number of stimuli since m is used for the dimension of relevant subspace, “we systematically this” (line 85); line 128, the dimensions are different on the two sides of the equation $WW^T = I_m$.

---

> ### Author Rebuttal · Authors · 2025-07-30
>
> We appreciate the reviewer’s favorable evaluation of our work. Despite the mentioned limitations, we believe our work sheds light on some novel and non-trivial aspects of representational dynamics under continual learning, which emerged even under simple setups. These are due to the nonlinear aspects of the learning signal and the associated dynamics in the parameter space, which is known to exist even in linear networks. To our knowledge, our work is also the first to theoretically study drift under different architectures and learning rules, pointing out the commonalities and differences specific to each setup. This line of research can certainly be expanded upon in future studies. We also thank the reviewer for pointing out the typo in line 128. The correct expression for the weight matrix is $W \in R^{m \times n}$.

---

> > ### Comment · Reviewer_DyoC · 2025-08-06
> >
> > I want to thank the authors for addressing my minor concerns, and I maintain my favorable rating.

---

> > > ### Author Response · Authors · 2025-08-06
> > >
> > > We greatly appreciate the feedback and the favorable rating provided by the reviewer.

---

### Official Review · Reviewer_6Soi · 2025-07-02

**Clarity:** 3
**Significance:** 3
**Originality:** 3
**Rating:** 4
**Confidence:** 3

**Summary:**

Representational drift occurs in neural networks, yet its mechanisms remain unclear. This study investigates how task-irrelevant stimuli contribute to drift in neural representations of task-relevant inputs, even when these irrelevant stimuli are learned to be ignored. Through theoretical analysis and simulations across various architectures including Hebbian-based learning and stochastic gradient descent models, the research demonstrates that drift rate increases with both the variance and dimensionality of task-irrelevant data. These findings establish important connections between stimulus structure, task context, and learning mechanisms in representational drift, potentially offering new approaches to understand underlying neural information processing in the brain.

**Questions:**

1）Please further explain the rationale for using Eq. 2 to define the first m dimensions as task-relevant and the remaining n-m dimensions as task-irrelevant. Specifically, what kind of neural representation or feature is this "task" intended to learn?

2）Please explain why in Eq. 2 all task-relevant dimension feature values are uniformly set to 1, while all task-irrelevant dimension feature values are set to the same constant less than 1?

3）It is suggested to add the related works in the introduction about the existing contributions of task-irrelevant stimuli in continuous learning tasks, and clearly articulate the innovation points of this research.

4）Regarding section 4.2, please elaborate on the task setting: the input consists of x1, x2, and n-2 dimensions of task-irrelevant Gaussian white noise, while the output is x1 and x2. How does this training method form receptive fields in the hidden layer? What is the mechanism? What role do task-irrelevant signals play, especially in reorganizing receptive fields?

**Ethical Concerns:**

["NO or VERY MINOR ethics concerns only"]

**Final Justification:**

I have no major concerns with this work and believe it's a interesting contribution to the field. I support its acceptance.

**Limitations:**

Please see the weaknesses and questions.

**Quality:**

3

**Strengths And Weaknesses:**

***Strengths***:
This paper employs theoretical analysis and computational simulation methods to study how task-irrelevant stimuli promote neural representation drift during continuous learning, with solid results.

***Weaknesses***:

1) The theoretical analysis results in this paper are only applicable to linear networks, and the input-output functions can only be linear, having certain limitations.

2) Both sections 4.2 and 4.3 indicate that task-irrelevant signals can cause neural representation drift during the learning process. What impact does this drift have on online learning performance?

---

> ### Author Rebuttal · Authors · 2025-07-30
>
> We appreciate reviewer’s insightful feedback on our work. Please find our responses to specific criticism below.
>
> 1. **Linear networks:**
> We agree that our theoretical derivations were limited to linear networks. However, we would like to note that the learning dynamics we analyzed are highly nonlinear functions of the parameters. We believe that the fact that both SGD and Hebbian-based learning lead to drift of parameters due to task-irrelevant stimuli is a consequence of nonlinearity in the learning dynamics that is already captured in linear networks. Although extending the analytics to networks with nonlinear activations would be an interesting avenue for future work, we do not believe it changes the main findings. This is exemplified by the numerical results in the nonlinear network in Section 4.2.
>
> 2. **Impact on online learning performance:**
> That’s a great question. Our main finding is that even after minimum loss is achieved, the presence of task-irrelevant stimuli can perturb the network, potentially leading to overall drift. In general, this effective noise might provide implicit regularization effects, causing  the network to favor specialized solutions. This effect may be observed in generalization performance. In our setup, while the average online performance is fixed, short-term fluctuations in performance increase as a function of the extent of task-irrelevant stimuli.
>
> 3. **Questions regarding spectrum in Eq.2:**
> In this section, we aimed to demonstrate the phenomenon in a simplest setup where the amount of task-irrelevant stimuli can be directly controlled. This design of the input covariance allows Oja’s network with input dimension $n$ and output dimension $m$ to learn to represent the principal subspace at the output. Here, the principal subspace corresponds to the subspace associated with $\lambda_{||}=1$, and any stimuli associated with $\lambda_{\perp}<1$ are task-irrelevant and vanish at the output layer. We show that even in this simple setup, the amount of drift increases with $\lambda_{\perp}$. In the later sections, we solve this for an arbitrary input covariance; hence, this motivating example serves as a special case of those results.
>
> 4. **Literature on contribution of task-irrelevant stimuli to continuous learning:**
> We thank the reviewer for this suggestion. We will add additional references related to continual learning, which often involves the flexible learning of multiple tasks in sequence without forgetting. Our finding is novel in that, even for a fixed task, the extent of ignored (task-irrelevant) stimuli affect the internal representations. This is applicable to many lifelong learning scenarios, where the agent is continuously learning from the environment. Understanding the nature of representation stability in these cases allows for better comparison of representations throughout learning and has important implications for areas such as interpretability research.
>
> 5. **Explanation of the nonlinear network:**
> We are happy to provide clarifications on this setup. With this example, we aimed to demonstrate that our main findings translate to a network with nonlinear activations. The input essentially consists of a task-relevant component, which denotes the position of an agent on a ring, as well as additional stimuli that are not relevant to the specific circuit (here represented by Gaussian stimuli). In general, the network can be viewed as performing computation on the task-relevant position variables. To accomplish this, the neurons must faithfully represent position on the ring in the middle layer, from which the downstream output is read out. Since neuron activations are non-negative, this leads to formation of localized receptive fields that tile the entire ring. We show that continuous training leads to repositioning of the receptor fields such that the tiling pattern and overall representation similarity are preserved (Fig 4b). Importantly, the extent of this depends on the extent of task-irrelevant stimuli. This is because the presence of those stimuli induces perturbations in learning, which can displace the system on the manifold of solutions and lead to a rearrangement of receptive fields. We would also like to note that, in the example, we chose reconstruction of position as the task, which causes the network to behave similarly to unsupervised non-negative similarity matching networks, such as ones seen in Ref. However, since our setup is supervised learning, the mapping on the task-relevant position variables can be chosen to be arbitrary, and we still see the same dependency. We are happy to provide additional plots along this line.
>
> Ref: Qin, S., Farashahi, S., Lipshutz, D. et al. Nat Neurosci 26, 339–349 (2023).

---

> > ### Comment · Reviewer_6Soi · 2025-08-05
> >
> > Thank you, these comments address my concerns.

---

> > > ### Author Response · Authors · 2025-08-05
> > >
> > > We are glad that our response addressed your concerns and greatly appreciate your feedback on our work.

---

### Official Review · Reviewer_RtGY · 2025-07-03

**Clarity:** 3
**Significance:** 2
**Originality:** 4
**Rating:** 4
**Confidence:** 3

**Summary:**

- The paper aims to understand the mechanisms of representational drift from a dual perspective of Hebbian and SGD-based learning.
- The authors present theory and experiments in both settings to show that task-irrelevant stimuli, which agents learn to ignore, can be sufficient to create long-term drift of representations of task-relevant stimuli.
- Through a motivating example of Hebbian-based learning, the authors show that drift rate increases with the variance and dimensionality of the data in the task-irrelevant subspace.
    - The authors show similar results using a non-linear network and SGD-based learning in the experiments section.
- With the assumption of continuous-time dynamics, the paper considers that a point on the solution manifold can be decomposed into local normal and tangential spaces. The authors particularly focus on late phase learning where the dynamics in the tangential space are purely diffusive.
- The authors derive the analytic form of angular diffusion in terms of task-irrelevant subspace properties in four settings: multidimensional Oja's network, similarity matching network, autoencoder, and a two-layer network.
- Using Gaussian toy data and MNIST, the authors show that drift rates closely match those predicted by the paper's theory.
- Finally, the authors contrast the dynamics under learning noise and external synaptic noise, showing that they have different characteristics.

**Questions:**

- Can a more general class of tasks be reduced to the problem of principal subspace tracking? If so, what is this class of tasks and how general is it?
- Beyond the observation that the two sources of noise have different impacts on the drift rate, are there any other takeaways from Section 5 comparing learning noise and synaptic noise?
- Typography (no impact on score)
    - Line 85: missing word: "we systematically ___ this"
    - Line 94: missing space before $\Delta$.

**Ethical Concerns:**

["NO or VERY MINOR ethics concerns only"]

**Final Justification:**

The authors engaged in the discussion to clarify some of the points raised in my review. Two notable elements of this discussion are summarized below:

- During the discussion, the authors provided a clearer definition of representational drift. I suggest the authors use a more formal definition to simultaneously introduce some of the notation they use in the paper's analyses.
- One of my main concerns was that the notion of "task" is discussed very generally throughout the paper, but the analysis presented is limited to the specific task of principal subspace tracking. In general, depending on the task, task-irrelevant stimuli can have higher variance and dimensionality. The authors explained how the supervised learning analysis in Appendix E can be presented more generally, where the task-relevant components are not necessarily the principal components. They also plan to include empirical analysis for this more general notion of "task."

Although I don't see a strong reason why the general messages regarding late phase learning representation drift would cease to hold in practical extended settings, the overall limited theoretical and empirical setup of the paper is still a concern when determining significance.

Based on these points, I increase my score to 4 (Borderline accept).

**Limitations:**

The paper does not discuss the limitation that it only considers the task of principal subspace tracking.

**Quality:**

2

**Strengths And Weaknesses:**

### Strengths

- Quality
    - The paper is motivated by citing relevant work to show that a similar notion of representational drift exists in neuroscience and machine learning.
    - Both Hebbian and SGD-based learning settings are studied to support the generality of the presented principles.
- Clarity
    - The paper is mostly clear and easy to follow.
- Originality
    - The analysis performed in the paper is novel.

### Weaknesses

- Quality
    - The central object of this paper, the notion of representational drift, needs to be more explicitly and rigorously defined, at least in the context of machine learning. Currently, part of this definition has to be inferred from the theoretical and experimental setups.
    - The notion of "task" is discussed very generally throughout the paper. However, a majority of the analysis is specific to the task of principal subspace tracking, so the generality of the notion of task-relevance and task-irrelevance is not supported. In general, depending on the task, task-irrelevant stimuli can have higher variance and dimensionality.
        - If more general tasks can be reduced to the studied problem of learning principal subspaces, this reduction procedure needs to be described.
- Significance
    - The limited theoretical and empirical setup of the paper makes it difficult to evaluate its significance for the community.

---

> ### Author Rebuttal · Authors · 2025-07-30
>
> We greatly appreciate the reviewer’s thorough assessment of our work. Please find responses to specific criticisms below.
>
> 1. **Generality of the results and the notion of task-irrelevancy:**
> We agree with the reviewer that in a great portion of the paper we used principal subspace tracking as the running task – this enabled us to study drift under different architectures and learning rules for the same task, which was a novel contribution of our work. However, as evidenced in the supervised learning setup, our work goes beyond principal subspace tracking task. In this setting, the task is determined by the teacher with $y = Px$, where $P$ is an arbitrary mapping from input to output. Here, task-irrelevant stimuli are those that lie in the null-space of P, for which the associated $y$ is $0$ (notably, this definition is consistent with the principal subspace task). In the two-layer supervised network presented in the main text, we intentionally designed $P$ to represent the principal subspace to provide a fair comparison with other networks. However, in the calculations provided in Appendix E for this network, there is no assumption restricting $P$ to represent the principal subspace. Indeed, the results hold for a more general mapping where the task-relevant subspace can be any arbitrary subspace of the input. In this case, the task-irrelevant induced drift appears as an extra term that is $\propto (n-k)\lambda_{\perp}$, where $k$ is the rank of $P$ (which in general could be less than the output dimension $m$), and $\lambda_{\perp}$ is the variance in the task-irrelevant dimension (i.e. the null-space of $P$). Importantly, $\lambda_{\perp}$ can have a variance higher than that of the task-relevant subspace. We are happy to clarify this point and add new text and plots to further corroborate this result. We thank the reviewer for this comment, which allows us to better clarify the scope of our work.
>
>
> 2. **Comparison of learning noise and synaptic noise:**
> The key point that we aimed to emphasize in Section 5 is that the learning noise can lead to qualitatively different drift compared to synaptic noise, even when all other parameters are held constant. This is because, for a given network, the subset of solutions that end up being explored depends on the dynamics of learning and the type of noise present during it. Additive Gaussian synaptic noise – which is a common way of inducing drift in the literature –  creates an isotropic noise profile in the parameter space. In contrast, the profile associated with learning noise can be anisotropic and position dependent in the parameter space. In our setup with Oja’s network, this distinction manifests not only in the overall rate of drift for a given stimulus, but also in the geometry of drift within the representation space. By this, we mean the relative rates at which representations of different stimuli drift along each other. We see that in Oja’s case, the synaptic noise leads to isotropic drift in the representation space with a pairwise drift rate of $D_{sr}=\eta\sigma_{syn}^2/4$, that is direction-independent (see line 225). By contrast, the corresponding learning noise results in variable drift rates between representations of two stimuli, depending on the input spectrum (see Eq.6). The difference in the drift geometry, and not only the overall drift rate, may be another experimentally measurable aspect of drift that can distinguish these two sources of noise. We thank the reviewer for prompting us to clarify this point.
>
>
> 3. **Definition of representational drift:**
> We share the reviewer’s sentiment regarding the need for a clear and widely accepted definition of representational drift, both in neuroscience and machine learning. In our work, we take drift to be simply the change in the internal representations of the network that occur without any obvious change in behaviour or task performance. In this context, it can refer to a late-stage learning where the performance (loss) is fixed on average. However, one can also imagine late stages of learning where certain types of solutions are gradually favored over others (e.g. slow transition to flatter or more generalizable solutions). In these situations, even though the training loss is not changing, the generalization may be improving.

---

> > ### Comment · Reviewer_RtGY · 2025-08-01
> >
> > Thank you for your response.
> >
> > I appreciate the authors' answer to my question regarding Section 5, it adds clarify to the general message of the section.
> >
> > > In our work, we take drift to be simply the change in the internal representations of the network that occur without any obvious change in behaviour or task performance. In this context, it can refer to a late-stage learning where the performance (loss) is fixed on average.
> >
> > Thank you. I think it would be beneficial to mention this explicitly up front to improve the paper's clarity. In fact, defining it more formally within the paper's scope can not only improve clarity, but also presents an excellent opportunity to gently introduce some of the notation used in the analyses. What does such a definition look like?
> >
> > > However, in the calculations provided in Appendix E for this network, there is no assumption restricting $P$ to represent the principal subspace. Indeed, the results hold for a more general mapping where the task-relevant subspace can be any arbitrary subspace of the input. In this case, the task-irrelevant induced drift appears as an extra term that is $\propto (n-k)\lambda_{\perp}$, where $k$ is the rank of $P$ (which in general could be less than the output dimension $m$), and $\lambda_{\perp}$ is the variance in the task-irrelevant dimension (i.e. the null-space of $P$). Importantly, $\lambda_{\perp}$ can have a variance higher than that of the task-relevant subspace.
> >
> > I took a look at Appendix E, but the analysis here appears to still be for the case of $P = I_{m,n} V^T$ s.t. $\Sigma_x = V \Lambda V^T$. Can the authors please indicate the relevant lines in the appendix that go beyond the principal subspace tracking problem? If this is not yet described, can you please elaborate the result and its interpretation for the more general notion of "task"?

---

> > > ### Author Response · Authors · 2025-08-04
> > >
> > > We truly appreciate the reviewer’s engagement with our work, and are happy that our response regarding Section 5 was satisfactory. Please find responses to the other points below.
> > >
> > > **On the definition of drift:** We welcome this suggestion and will ensure that the definition is clarified up front in the paper. Since we are using online learning, the notion of fixed loss in our case means that if $L = \langle l \rangle_{data}$ is the expected loss over the data distribution, then $L$ should reach a steady state over time (in practice, this can be verified by placing a percentage threshold on the rate of change). Note that for the Hebbian networks, where there’s no explicit loss defined, we instead use a surrogate loss (specifically, the Grassmann distance, which quantifies the distance to the principal subspace). Additionally, since the changes to the representations are caused by a diffusion process, and given that the norms of the representations are fixed after steady state, we quantify drift by measuring pairwise angular diffusion rates for representations of a given set of trial stimuli. Importantly, the total diffusion rate is proportional to the rate of decay (slope) of autocorrelation plots over time -- a measure that is commonly used in experimental setups to quantify drift.
> > >
> > > **Clarifications regarding the supervised task:** The reviewer is correct that in Appendix E we assumed the teacher mapping is $P = I_{m,n}V^T$. This, by design, represents the top $m$ rows of $V^T$ at the output. Below we explain in more detail why the fact that $P$ represents the principal subspace is not essential to these calculations, and hence, the supervised network can represent more general tasks without changing the main findings. We apologize if this was not clear from our previous response.
> > >
> > > Consider a mapping that, instead of the principal subspace, represents $m$ arbitrary rows of $V^T$ at the output (i.e. "task-relevant" subspace). This mapping can be denoted by $P = I_{m,n}\bar{V}^T$, where $\bar{V}^T$ is a row-shuffled version of $V^T$ such that its first $m$ rows constitute the task-relevant subspace. If we assume the stimuli covariance in the task-relevant has eigenvalue $\lambda_{||}=1$, and in the task-irrelevant subspace $\lambda_{\perp}$, this essentially becomes identical to calculations in Appendix E, except $V$ needs to be substituted with $\bar{V}$ from Eq. 66 onwards. This is because the effective covariance $\Sigma'$ in Eq.65 still remains the identity, and the solutions of the manifold (Eq. 66) and the subsequent Hessian essentially remain unchanged, except for substitution of $V$ with $\bar{V}$. Similarly, the final results in Eq. 76 also remain intact, except that the indices in the input space are now sorted based on rows of $\bar{V}$). Note that there was no restriction on the value of $\lambda_{\perp}$ and it could be greater than 1.
> > >
> > > The above case corresponds to a full-rank $P$. If $P$ indeed has rank $k < m$, there will be a slight change to the derivations above, since now $m-k$ rows of $P$ are zero. This gets reflected in the change of variables to ”primed” coordinates in Eq. 64 and the associated reduced network in Eq. 65 such that $m$ needs to be replaced with $k$ (this means the reduced “primed” weights now have dimensions $k \times p$ and $p \times k$, which is compatible with the fact that the solution weights have rank $k$). This change is manifested in the final drift equations only by replacing $m$ with $k$.
> > >
> > > What is clear from these calculations, and those for the other networks, is that the additional drift term due to the task-irrelevant stimuli stems not from the specifics of the task and the associated objective, but the learning-induced noise that can perturb the system outside the task-dependent manifold of solutions. Interestingly, these perturbations corresponds to the subspace $n_1$ of the Hessian, which emerges with a similar structure for all the networks, irrespective of the particular task and learning rule. Hence, we believe this is a more general phenomenon and goes beyond principal subspace tracking.
> > >
> > > **Additional results:**
> > > We are happy to provide new results and calculations that demonstrate the point that our supervised task applies to more general mappings. This includes augmenting/revising Appendix E along the above points, as well as new experimental results that corroborate these points. Specifically:
> > >
> > > 1) A plot showing that for arbitrary $P$ (chosen randomly from a Gaussian distribution), increasing $\lambda_{\perp}$ in the task-irrelevant space (i.e. the null-space of $P$) leads to higher drift,
> > > 2) A similar plot showing that, when $P$ is low-rank, drift rate depends on $k=rank(P)$ and not the output dimension $m$.
> > >
> > > We thank the reviewer again for their feedback, which allowed us to clarify these points and improve our work. We are more than happy to answer any further questions.

---

> > > > ### Comment · Reviewer_RtGY · 2025-08-04
> > > >
> > > > Thank you for your detailed response; it answers my questions and addresses my concerns regarding the definition of drift and the generality of "task" as used in the paper.

---

> > > > > ### Author Response · Authors · 2025-08-05
> > > > >
> > > > > We are pleased that our response addressed reviewer's questions and concerns, and are grateful for your constructive criticism of our work.

---

### Official Review · Reviewer_eXNF · 2025-07-03

**Clarity:** 3
**Significance:** 3
**Originality:** 3
**Rating:** 5
**Confidence:** 4

**Summary:**

In this article, the authors analytically study drift of neural representation under continual learning of stimuli with 'non-relevant' directions in linear networks. They provide a general framework to study and quantify the phenomenon, and apply it to a variety of setups. They find good agreement with numerical simulations, and differentiate the resulting dynamics from drift induced by explicit noise added to the weights.

**Questions:**

See above.

**Ethical Concerns:**

["NO or VERY MINOR ethics concerns only"]

**Final Justification:**

My points were addressed, I believe the results of this paper, while limited in scope, are novel and interesting. I recommend acceptance.

**Quality:**

3

**Strengths And Weaknesses:**

## Strengths
1. The article addresses a relevant question, and provides explicit and understandable results. The selection of learning setups is sensible and comprehensive.
## Minor Weaknesses
2. When introducing the local coordinates for the main SDE (4), the authors could improve the clarity by clearly explaining what the expansion point is, and being more explicit about the assumption underlying the equations.
3. The results for non-linear models could be extended by investigating how the input spectrum determines the drift rate, and how this compares to the linear case.
4. Representational drift and its geometry has been studied and characterized in various cortical areas. A brief discussion on which aspects of the experimental observations are consistent with the effects seen in the model and which ones are not would be a valuable addition.

---

> ### Author Rebuttal · Authors · 2025-07-30
>
> We appreciate the reviewer’s positive assessment of our work. Please find specific responses below.
>
> 1. **Clarifications on SDE of Eq.4:**
> We are happy to provide clarifications regarding the assumptions underlying this set of equations. First, these equations are based on approximating the discrete learning update by continuous-time stochastic differential equations, which has been widely used in the context of SGD (e.g. Ref. 1). The main assumption behind this approximation is a small learning rate. We apply this approximation around a stable point on the manifold of solutions, $\tilde{\theta}$, where the average loss is zero and the Hessian spectrum is non-negative. When there is an explicit loss associated with the network, the Hessian becomes symmetric and can be block-diagonalized into subspaces with positive and zero eigenvalues. As a result, the dynamics can be decomposed into normal and tangent spaces leading to fast and slow dynamics (e.g. Ref. 2). We found Hessian to be symmetric in all the models except Similarity Matching (for which there’s no explicit loss function). However, we were able to perform a similar decomposition of dynamics using non-orthogonal projections (see Appendix C). Furthermore, our calculation of drift in the tangent space is based on the assumption of small fluctuations around the manifold, which allows us to calculate the diffusion into the tangent space by integrating the first order term of the deviation. When the deviation is large from the manifold, one needs to include contributions from second and higher order terms to the drift.
>
> 2. **Dependency on the spectrum in nonlinear case:**
> Thank you for this suggestion. We believe that deriving a closed-form equation for the drift as a function of the input spectrum is in general challenging for nonlinear networks. This is mainly because there are often no closed-form solutions in the weight space for nonlinear networks. However, given such solutions, the same methodology could be applied to derive the drift. As was demonstrated in the nonlinear network example, the general relationship – that drift increases as a function of the spectrum’s magnitude in the task-irrelevant subspace– holds. However, the exact dependence on the spectrum may depend on the type of solution amongst other parameters. We would also like to note that even in our linear cases, the theoretical dependencies turned out to be rather complicated in terms of the whole spectrum (see e.g. Eqs 6, 8 and 9), and only simplified under more structured input spectra.
>
> 3. **Relation to cortical drift in experiments:**
> This is an excellent question. Even though drift has been observed across various cortical areas and under different experimental designs, we did not find any studies that specifically examined the influence of task-irrelevant stimuli. However, we can speculate about potential connections to some existing findings. In one related study (Ref 3), the amount of drift observed in an association area – where different types of stimuli need to be multiplexed – appears to be higher than the drift at the origin areas. This might be related to the phenomenon discussed in our study as different sets of stimuli are combined with varying levels of relevance depending on the task.  Additionally, a potentially related observation comes from the visual cortex, where more complex and naturalistic stimuli show higher drift compared to simpler stimuli (Ref 4). This could also relate to our finding that, in a network with a limited representational capacity (such as the bottleneck in our work), a more complex stimulus is effectively equivalent to activating a higher number of input neurons, resulting in a greater amount of “task-irrelevant” content, and consequently a higher drift. As mentioned in the discussion, a more controlled test would be a setup in which the amount of distracting background stimuli can be systematically varied. This could be done, for example, in a vision task of the type used in the first study above, or in an olfactory target detection task with varying levels of background stimuli.
>
> References:
>
> Ref 1: Stephan Mandt, Matthew D Hoffman, and David M Blei. arXiv preprint arXiv:1704.04289, 2017.
>
> Ref 2: Zhiyuan Li, Tianhao Wang, and Sanjeev Arora. ICLR, 2022.
>
> Ref 3:  Franco, L.M., Goard, M.J. Nat Commun 15, 6872 (2024).
>
> Ref 4: Marks, T.D., Goard, M.J. Nat Commun 12, 5169 (2021).

---

> > ### Comment · Reviewer_eXNF · 2025-08-04
> >
> > Thank you for your detailed response. Your clarifications satisfactorily addressed the questions I had. I encourage you to incorporate the proposed clarification and the new references into the camera-ready version of the paper. I maintain my positive assessment and recommend acceptance.

---

> > > ### Author Response · Authors · 2025-08-05
> > >
> > > We are glad that our clarifications were satisfactory. We will certainly incorporate the new information into the updated version of the paper. Thank you again for your valuable feedback on our work.

---

### Decision · Program_Chairs · 2025-09-17

**Decision:**

Accept (poster)

**Comment:**

The recommendation is to accept.

This paper presents a novel and technically sound analysis showing that task-irrelevant stimuli are a significant source of representational drift. The reviewers reached a positive consensus, praising the paper's originality and the strong agreement between theory and simulation across different learning paradigms. The author-reviewer discussion was particularly effective. The authors provided convincing responses to initial concerns, especially regarding the generality of the "task," clarifying that the findings extend beyond principal subspace tracking.

For the camera-ready version, it's essential that the authors incorporate the changes promised in the rebuttal. Specifically, please add the clearer definition of representational drift upfront, expand on the generality of the task framework, and include the discussion connecting your findings to experimental observations of cortical drift.